# Serotonin and dopamine modulate aging in response to food odor and availability

Hillary A. Miller [1,4], Shijiao Huang [2,4], Elizabeth S. Dean [2], Megan L. Schaller[2], Angela M. Tuckowski[1], Allyson S. Munneke[1], Safa Beydoun [2], Scott D. Pletcher[2] & Scott F. Leiser [2,3✉]

An organism's ability to perceive and respond to changes in its environment is crucial for its health and survival. Here we reveal how the most well-studied longevity intervention, dietary restriction, acts in-part through a cell non-autonomous signaling pathway that is inhibited by the presence of attractive smells. Using an intestinal reporter for a key gene induced by dietary restriction but suppressed by attractive smells, we identify three compounds that block food odor effects in *C. elegans*, thereby increasing longevity as dietary restriction mimetics. These compounds clearly implicate serotonin and dopamine in limiting lifespan in response to food odor. We further identify a chemosensory neuron that likely perceives food odor, an enteric neuron that signals through the serotonin receptor 5-HT1A/SER-4, and a dopaminergic neuron that signals through the dopamine receptor DRD2/DOP-3. Aspects of this pathway are conserved in *D. melanogaster*. Thus, blocking food odor signaling through antagonism of serotonin or dopamine receptors is a plausible approach to mimic the benefits of dietary restriction.

[1] Cellular and Molecular Biology Program, University of Michigan, Ann Arbor, MI 48109, USA. [2] Molecular & Integrative Physiology Department, University of Michigan, Ann Arbor, MI 48109, USA. [3] Department of Internal Medicine, University of Michigan, Ann Arbor, MI 48109, USA. [4] These authors contributed equally: Hillary A. Miller, Shijiao Huang. ✉email: leiser@umich.edu

Rapid advances in aging research have identified several conserved signaling pathways that influence aging in organisms across taxa[1]. Recent work shows that many of these "longevity pathways" act through cell non-autonomous signaling mechanisms[2,3]. These pathways utilize sensory cells—frequently neurons—to signal to peripheral tissues and promote survival during the presence of external stress. Importantly, this neuronal activation of stress response pathways, through either genetic modification or exposure to environmental stress, is often sufficient to improve health and longevity. Despite mounting evidence that neuronal signaling can influence multiple longevity pathways, less is known about the cells and molecules that propagate these signals.

Biogenic amines are among the most well-studied and conserved neuronal signaling molecules[4,5]. Specifically, serotonin and dopamine play well-defined roles in behavior and physiology. However, their role in aging is less well understood. Several recent studies implicate serotonin as an important signal in multiple C. elegans longevity pathways including the response to heat shock[6], hypoxia[7] and dietary restriction (DR)[8,9]. tph-1 (WormBase, ZK1290.2), the rate-limiting enzyme for serotonin synthesis, is involved in food abundance sensing and regulates food deprivation-mediated longevity through both the ADF and NSM neurons[8]. Perception of food smell dampens the longevity benefit of DR[10–12]. A recent report also shows that tph-1 knockout mutants do not respond to the food smell suppression of DR-mediated longevity, and ADF neuron activity is involved in response to food smell[9]. Dopaminergic signaling is associated with physical activity in humans and loss of this signaling decreases lifespan in mice[13] and blocks lifespan extension in nematodes[14]. Serotonin and dopamine levels both decrease with age across species[15,16], consistent with these signaling pathways promoting healthy aging. A pharmacological screen for extending C. elegans lifespan has identified that compounds modulating serotonin and dopamine signaling promote longevity[17]. Despite rigorous study and clinical use of drugs that modify serotonin and dopamine signaling, our understanding of their complex actions and potential interaction is far from complete.

Dietary restriction (DR) is the most well-studied and consistent intervention known to improve health and longevity in organisms ranging from single-celled yeast to primates[18]. DR leads to improved cell survival and stress resistance, complex intracellular signaling events, metabolic changes, and increased activity in multiple organisms. Nematode flavin-containing monooxygenase-2 (encoded by fmo-2, WormBase, K08C7.5) is necessary and sufficient to increase health and longevity downstream of DR[7]. FMOs are highly conserved proteins that are also induced in multiple mammalian models with increased lifespan, including DR[19,20]. Having previously identified a role for fmo-2 in aging, we wondered whether DR cell non-autonomously regulates fmo-2 induction and whether perception of food through biogenic amines could be involved in the subsequent signaling pathway.

In this work, we use fmo-2 induction to interrogate the cell non-autonomous DR signaling pathway and how it is affected by food odor. We find that DR induces fmo-2 cell non-autonomously to increase longevity through a pathway that involves decreased serotonin signaling from NSM serotonergic neurons and decreased dopamine signaling from dopaminergic neurons. This process is abrogated by food odor sensed by AWC chemosensory neurons and can be mimicked through small molecules that antagonize biogenic amine signaling. Lastly, we find that some of these processes are conserved in fruit flies, suggesting that blocking the perception of food is a plausible approach to mimicking DR.

## Results

**Attractant food odor represses fmo-2 to limit longevity**. We developed an integrated single-copy mCherry reporter driven by the fmo-2 promoter to measure fmo-2 induction. The reporter is primarily expressed in the intestine and responds to stimuli previously reported to induce fmo-2, including DR. As an intestinal protein[21], we expected that fmo-2 would likely be induced cell autonomously by the change in nutrient intake under DR. To test this hypothesis, we asked whether the perception of food smell by worms in the absence of eating can abrogate the induction of fmo-2. Using a "sandwich plate" assay as described in Fig. 1a, we were surprised to find a significant reduction in fmo-2 induction when worms could smell but not eat food (Fig. 1b, c, C. elegans strains used in this study are listed in Supplementary Data 1). This reduction is consistent with a model in which increased fmo-2 mediates the increase in longevity from DR, as food smell significantly abrogates lifespan extension by DR, (Fig. 1d, all lifespan replicates in this study are listed in Supplementary Data 2) similar to previous findings[9–12]. We also find that live bacteria are required to abrogate fmo-2 induction, as the presence of bacteria killed with 0.5% paraformaldehyde[22] does not prevent DR from inducing fmo-2 expression (Supplementary Fig. S1a, b). Since intestinal cells are not known to perceive external environmental cues such as smell, these results suggest that fmo-2 expression is suppressed when live food is present through cell non-autonomous signaling.

We next wondered what types of odorants worms sense in this pathway. Bacteria secrete hundreds of volatile compounds that are classified in three categories based on how they promote chemotaxis: attractants, repellants, and neutral compounds[23–25]. We tested whether exposure to any volatile compound secreted from bacteria is sufficient to block the lifespan-promoting effects of DR or whether compounds identified as attractants and repellants oppositely regulate fmo-2 induction. Using compounds derived from studies of the E. coli strain HB101 in a range of concentrations (Supplementary Data 3), we find that attractants are more likely to suppress DR-mediated induction of fmo-2 (Fig. 1e, f) whereas some neutral and repellant compounds can induce fmo-2 under fed conditions (Supplementary Fig. 1c–h). We also find that many attractive compounds suppress fmo-2 expression, consistent with the hypothesis that this pathway is not acting through a single receptor (Fig. 1g, all results in Supplementary Fig. 2a–z). These results support a model in which perception of attractive odorants secreted by E. coli abrogates the induction of the pro-longevity gene fmo-2. To test whether these odors also affect longevity, we exposed worms to one odorant from each category and measured lifespans +/− DR. Our results show that the attractant shortened DR lifespan only, recapitulating the effect of food smell on DR mediated lifespan; the neutral odorant did not affect lifespans of fed or DR worms; while the repellant shortened fed and DR lifespans (Fig. 1h, Supplementary Fig. 1i–j). This is consistent with attractive smells preventing the lifespan-promoting effects of DR, possibly through a neural response to external stimuli that leads to physiological changes in peripheral tissues.

**Serotonin and dopamine antagonists induce fmo-2 to mimic DR longevity**. Biogenic amines can regulate pro-longevity pathways and are involved in behavioral changes in response to food[6,7,26–28]. Some biogenic amines have previously been reported to regulate DR mediated longevity[8,9,17,29]. We next asked whether neurotransmitters are involved in the fmo-2-mediated food odor pathway. Using a targeted approach focusing on neurotransmitters and their antagonists, we tested for compounds sufficient to prevent the abrogation of fmo-2 induction in the

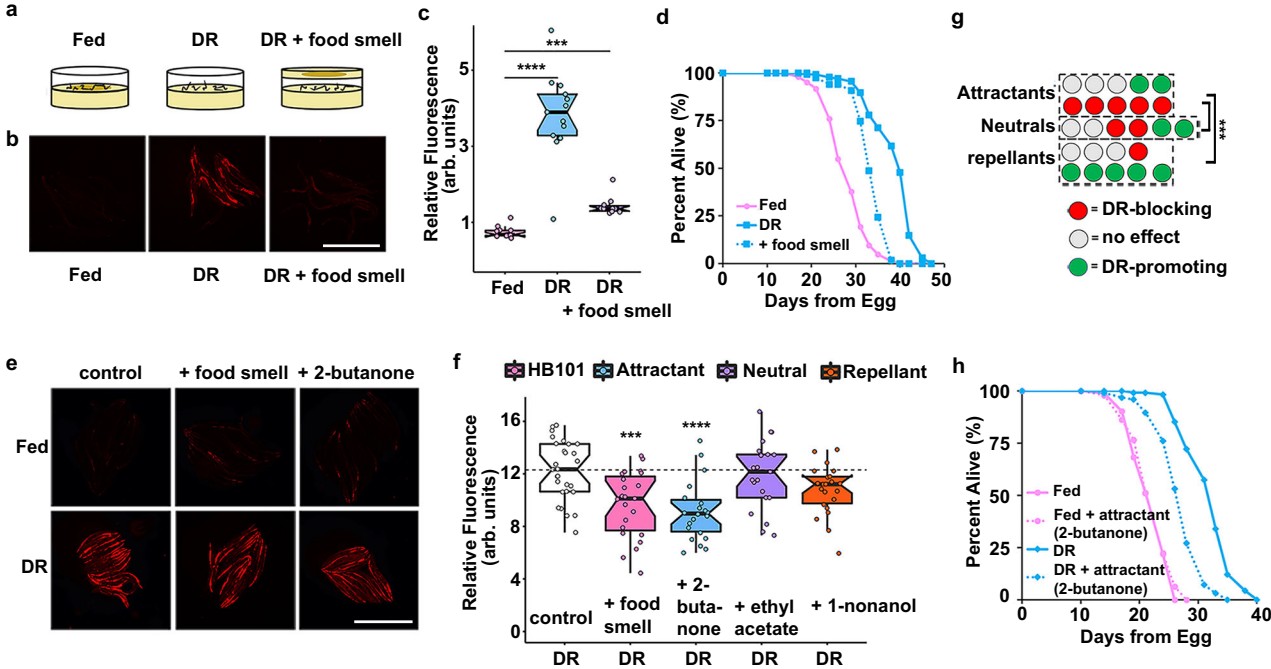

**Fig. 1 Attractive food smell blocks dietary restriction-mediated *fmo-2* induction and longevity.** Diagram of "smell plates" (**a**). Images (**b**) and quantification (**c**) of individual *fmo-2p*::*mCherry* worms on fed (pink), DR (blue) and DR + food smell (OP50) (purple). Scale bar, 1 mm. *n* = 12 (fed), 13 (DR), 10 (DR + food smell) biologically independent animals. *p*-value = 2.95e-09 (DR vs. fed), 2.14e-06 (DR + food smell vs. fed). Survival curves (**d**) of N2 (WT) animals fed (pink) or DR (blue) under normal conditions (solid lines) or subjected to the smell of bacteria (dotted line). Images (**e**) and quantification (**f**) of individual *fmo-2p*::*mCherry* worms on DR plates exposed to food smell (HB101) (pink) or attractive (2-butanone in blue), neutral (ethyl acetate in purple), or repellant (1-nonanol in orange) odorants. Scale bar, 1 mm. *n* = 28 (DR), 23 (DR + food smell), 22 (DR + 2-butanone), 25 (DR + ethyl acetate), 22 (DR + 1-nonanol) biologically independent animals. *p*-value = 0.000374 (DR vs. DR + food smell), 1.12e-05 (DR vs. DR + 2-butanone), 0.442 (DR vs. DR + ethyl acetate), 0.9993 (DR vs. DR + 1-nonanone). **g** Summary of the effects of 26 odorants on *fmo-2* induction during DR. Survival curves (**h**) of N2 (WT) animals fed (pink) or DR (blue) under normal conditions (solid lines) or subjected to attractive odorants (dotted line). \*\*\**P* < .001, \*\*\*\**P* < .0001 when compared to fed (Welch Two Sample t-test, two-sided). The box plots display the median by the middle line of the box. The upper boundary of the box indicates the 75% interquartile range, and the lower boundary indicates the 25% interquartile range.

presence of food smell (Supplementary Fig. 3a–d). The biogenic amine neurotransmitter antagonists mianserin (for serotonin) and thioridazine and trifluoperazine (for dopamine) consistently and significantly restore *fmo-2* induction to DR levels in the presence of food smell (Fig. 2a–c, Supplementary Fig. 3e, f). Mianserin is a tetracyclic serotonin antagonist that is thought to competitively bind to serotonergic G protein-coupled receptors (GPCRs)[30] while thioridazine and trifluoperazine's mechanism of action involves blocking dopamine receptors[31]. Importantly, while each compound induces *fmo-2* to a different extent (Fig. 2d, Supplementary Fig. 3g, i), when combined with DR, no antagonist further induced *fmo-2*, suggesting they act in the same pathway (Fig. 2e, Supplementary Fig. 3h). Diphenyleneiodonium chloride (DPI), an inhibitor of NADPH oxidase, which we identified in an unpublished screen to robustly induce *fmo-2* additively with DR, acts as a positive control, and further induces *fmo-2* when combined with DR (Fig. 2e). Because thioridazine and trifluoperazine act through similar mechanisms and the effects of thioridazine were more consistent in our studies, we focused further experiments on dopamine antagonism through thioridazine. Together, these results support antagonism of serotonin or dopamine as partial mimetics of DR in their induction of *fmo-2*.

To validate that the induction of *fmo-2* through biogenic amine antagonism is beneficial for longevity, we next asked whether these compounds extend lifespan. We find that both mianserin and thioridazine extend lifespan on agar plates in a dose-dependent manner (Fig. 2f–g). Previous studies report that

mianserin also extends lifespan in liquid culture[29], but were not replicated on solid agar plates[32]. We also confirmed that *fmo-2* is induced at 25 µM mianserin or 25 µM thioridazine (Supplementary Fig. 3j–k). Since we identified mianserin and thioridazine through their induction of *fmo-2*, and previously found that *fmo-2* is necessary for DR-mediated lifespan extension, we next asked whether *fmo-2* was necessary for the beneficial longevity effects of mianserin or thioridazine. Our results show that the *fmo-2* loss of function completely blocks the lifespan effect of mianserin (Fig. 2h) and thioridazine (Fig. 2i). Importantly, we also see that mianserin treatment combined with DR does not further extend lifespan (Supplementary Fig. 3l). These results are consistent with these compounds mimicking some aspects of DR-signaling, recapitulating part of the DR lifespan extension effect. Collectively, this supports a model where DR induces *fmo-2* because of decreased biogenic amine signaling and establishes neuromodulators as a useful tool to decipher where in the signaling pathway a cell, signal, or receptor plays a role in DR-mediated longevity.

**Odor sensing AWC neurons modulate food smell response.** Our initial results establish that antagonizing serotonin and dopamine signaling leads to induction of the longevity promoting *fmo-2* gene and rescue of the negative effects of food smell. Based on this, we hypothesized that the relative lack of food smell during DR leads to increased longevity through induction of intestinal *fmo-2*. Using this framework, we next sought to better understand how the sensing of bacteria (or lack thereof) is

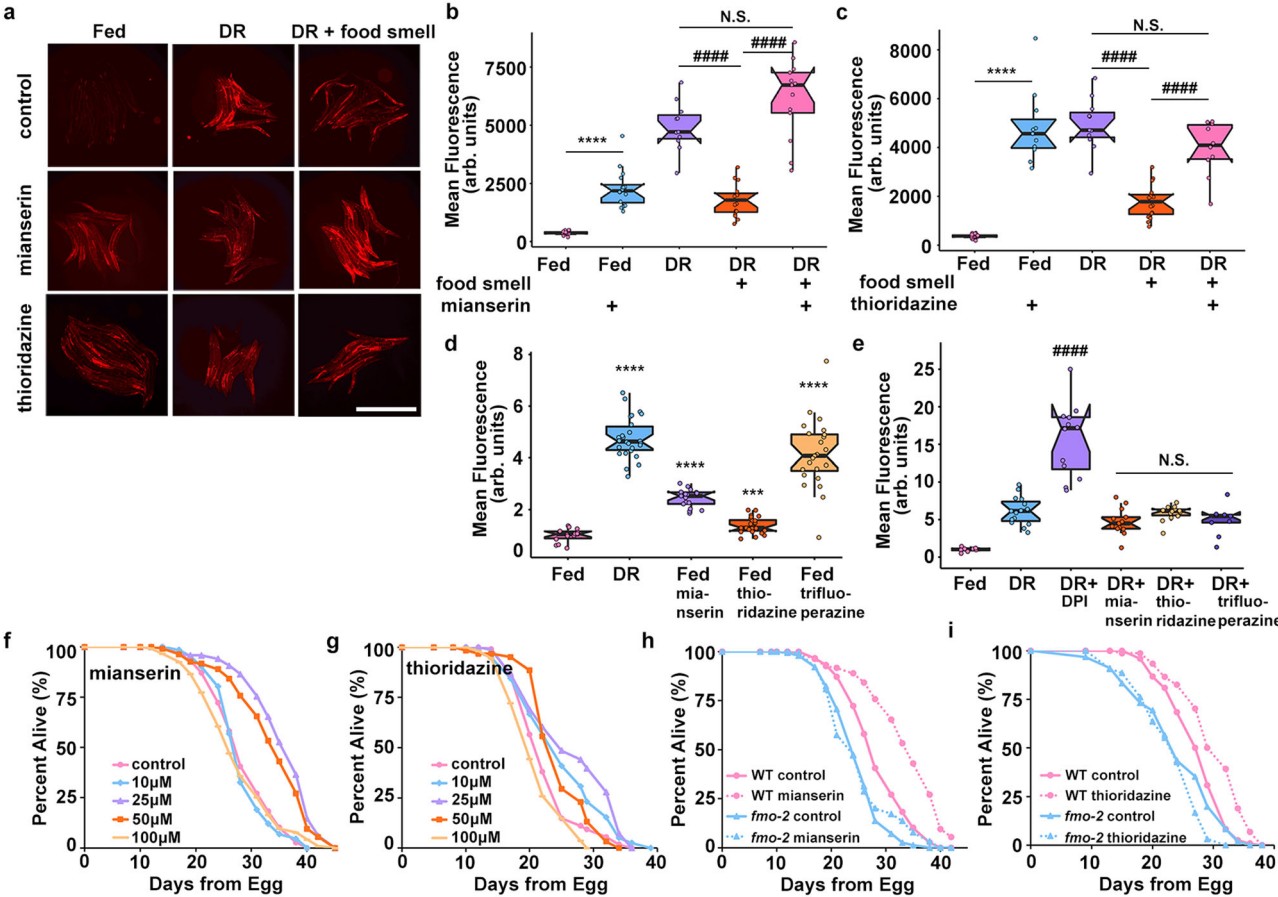

**Fig. 2 Serotonin and dopamine antagonists induce *fmo-2* and extend lifespan.** Images (**a**) and quantification of *fmo-2p::mCherry* given 100 μM of mianserin (**b**) or thioridazine (**c**) (blue) in combination with DR and food smell (pink) compared to DR alone (purple). Scale bar, 1 mm. *n* = 16 (fed), 14 (fed + mianserin), 11 (DR), 16 (DR + food smell), 13 (DR + food smell + mianserin), 11 (fed + thioridazine), 10 (DR + food smell + thioridazine) biologically independent animals. *p*-value = 3.45e-09 (fed vs. fed + mianserin), 9.77e-10 (DR vs. DR + food smell), 3.34e-10 (DR + food smell vs. DR + food smell + mianserin), 0.0533 (DR vs. DR + food smell + mianserin), 6.42e-12 (fed vs. fed + thioridazine), 2.01e-06 (DR + food smell vs. DR + food smell + thioridazine), 0.0515 (DR vs. DR + food smell + thioridazine). Quantification of Fed, DR, and DR + food smell in **b** and **c** are identical as data in **b** and **c** were acquired concurrently. Quantification (**d**) of *fmo-2p::mCherry* exposed to water (pink), DR (blue), 100 μM mianserin (purple), thioridazine (orange), or trifluoperazine (yellow). *n* = 18 (fed), 22 (DR), 17 (fed + mianserin), 24 (fed + thioridazine), 24 (fed + trifluoperazine) biologically independent animals. *p*-value = 1.82e-19 (fed vs. DR), 2.29e-13 (fed vs. fed mianserin), 0.000345 (fed vs. fed thioridazine), 1.73e-10 (fed vs. fed trifluoperazine). Quantification (**e**) of *fmo-2p::mCherry* exposed to water (pink) or DR (blue) in combination with 500 μM DPI (purple), 100 μM mianserin (orange), 100 μM thioridazine (yellow), or 100 μM trifluoperazine (dark purple). *n* = 8 (fed), 15 (DR), 13 (DR + DPI), 13 (DR + mianserin), 14 (DR + thioridazine), 9 (DR + trifluoperazine) biologically independent animals. *p*-value = 1.69e-06 (DR vs. DR + DPI), 0.0955 (DR vs. DR + mianserin), 0.633 (DR vs. DR + thioridazine), 0.136 (DR vs. DR + trifluoperazine). Survival curves (**f**) of N2 (WT) animals treated with 0 μM (water; pink), 10 μM (blue), 25 μM (purple), 50 μM (orange), or 100 μM (yellow) mianserin. Survival curves (**g**) of WT animals treated with 0 μM (water; pink), 10 μM (blue), 25 μM (purple), 50 μM (orange), or 100 μM (yellow) thioridazine. Survival curves (**h**) of WT animals (pink) and *fmo-2* KO animals (blue) on water (solid lines) or 50 μM mianserin (dotted lines). Survival curves (**i**) of WT animals (pink) and *fmo-2* KO animals (blue) on water (solid lines) or 25 μM thioridazine (dotted lines). ***P < .001, ****P < .0001 when compared to fed (Welch Two Sample *t*-test, two-sided). ####P < .0001 when compared to DR or DR + food smell (Welch Two Sample *t*-test, two-sided). The box plots display the median by the middle line of the box. The upper boundary of the box indicates the 75% interquartile range, and the lower boundary indicates the 25% interquartile range.

communicated to intestinal cells during DR. Our initial results, knocking down *unc-13*, a gene required for both synaptic vesicle[33] and dense core vesicle exocytosis[34], support short-range neurotransmitters and/or long-range neuropeptides as necessary for *fmo-2* induction (Supplementary Fig. 4a, b).

In *C. elegans*, perception of the external environment is largely regulated by a specialized organ known as the amphid. A previous report using a solid-liquid DR approach suggested a pathway originating in the ASI amphid neurons[35]. DAF-7/TGFβ produced by the ASI neurons modulates DR longevity[8,36,37]. We first asked whether these cells are required to modulate *fmo-2* activity during DR. We find that loss of *daf-3* (WormBase, F25E2.5) or *daf-7* (WormBase, B0412.2), each necessary for chemoreceptor

signaling in the ASI neurons, did not affect the food odor-mediated reduction in *fmo-2* expression (Supplementary Fig. 4c, d). Similarly, proper formation of the amphid (*daf-6*, WormBase, F31F6.5) is also not required (Supplementary Fig. 4e, f). However, it is reported that *daf-6* mutants can still respond to volatile odorants[23]. Three sensory neuron pairs, the AWA/B/C, do not possess the ciliated projections lacking in the *daf-6* mutant[23]. To test the necessity of these sensory neurons, we created individual genetic ablation strains by expressing the pro-apoptotic *caspase-3* gene (NCBI Gene, 836) under promoters of genes specifically expressed in AWA, AWB, or AWC neurons and crossed them into our *fmo-2p::mCherry* reporter. We find that loss of AWC neurons prevents *fmo-2* suppression in the presence of food smell

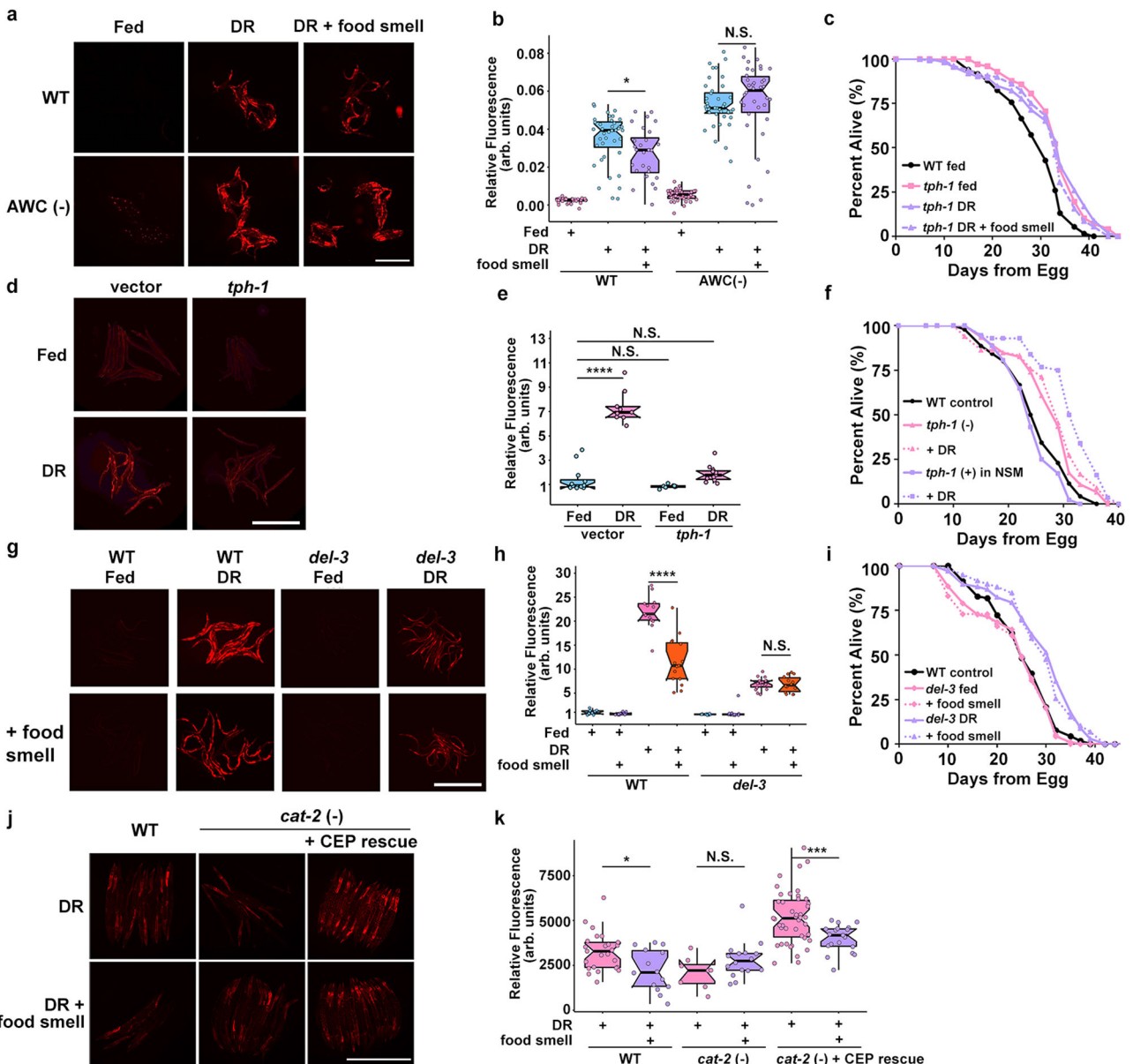

**Fig. 3 Food signals requires the AWC, NSM, and CEP neurons.** Images (**a**) and quantification (**b**) of *fmo-2p::mCherry* in a WT (control) and AWC genetic ablation background on fed (pink), DR (blue), or DR exposed to food smell (purple). Scale bar, 1 mm. n = 36 (WT fed), 40 (WT DR), 28 (WT DR + food smell), 41 (AWC(-) fed), 40 (AWC(-) DR), 38 (AWC(-) DR + food smell) biologically independent animals. p-value = 0.0203 (WT DR vs. WT DR + food smell), 0.926 (AWC(-) DR vs AWC(-) DR + food smell). Survival curves (**c**) of WT animals (black) and *tph-1* KO animals on fed (pink) and DR (purple) conditions exposed to food smell (dotted lines). Images (**d**) and quantification (**e**) of *fmo-2p::mCherry* exposed to *tph-1* RNAi on fed (blue) or DR (pink). Scale bar, 1 mm. *n* = 12 (vector fed), 9 (vector DR), 10 (*tph-1* fed), 11 (*tph-1* DR) biologically independent animals. *p*-value = 5.87e-10 (vector fed vs. vector DR), 0.108 (vector fed vs. *tph-1* fed), 0.221 (vector fed vs. *tph-1* DR). Survival curves (**f**) comparing control (black), *tph-1* KO (pink), and *tph-1* NSM-specific rescue (purple) animals on fed (solid line) and DR (dotted lines). Control (black) survival curve is also displayed in Supplementary Fig. 5F. Data in **f** and Supplementary Fig. 5f were acquired concurrently. Images (**g**) and quantification (**h**) of *fmo-2p::mCherry* in a WT (control) and *del-3* KO background on fed (blue) and DR (pink) exposed to food smell (purple and orange, respectively). Scale bar, 1 mm. *n* = 11 (WT fed), 13 (WT fed + food smell), 13 (WT DR), 15 (WT DR + food smell), 10 (*del-3* fed), 12 (*del-3* fed + food smell), 14 (*del-3* DR), 11 (*del-3* DR + food smell) biologically independent animals. *p*-value = 1.32e-06 (WT DR vs. WT DR + food smell), 0.212 (*del-3* DR vs. *del-3* DR + food smell). Survival curves of conditions comparing WT (black) to *del-3* (**i**) on fed (pink) and DR (purple) conditions in combination with food smell (dotted lines). WT control (black) survival curve is also displayed in Supplementary Fig. 6a. Data in **i** and Supplementary Fig. 6a were acquired concurrently. Images (**j**) and quantification (**k**) of *fmo-2p::mCherry* in a WT, *cat-2* KO, or *cat-2* KO with *cat-2* CEP neuron rescue on DR (pink) and DR + food smell (purple). Scale bar, 1 mm. *n* = 25 (WT DR), 14 (WT DR + food smell), 8 (*cat-2*(-) DR), 16 (*cat-2*(-) DR + food smell), 43 (*cat-2*(+) + CEP rescue DR), 19 (*cat-2*(-) + CEP rescue DR + food smell) biologically independent animals. *p*-value = 0.0103 (WT DR vs. WT DR + food smell), 0.0922 (*cat-2* (-) DR vs. *cat-2* (-) DR + food smell), 0.000114 (*cat-2* (-) + CEP rescue DR vs. *cat-2* (-) + CEP rescue DR + food smell). *P < .05, ***P < .001 and ****P < .0001 (Welch Two Sample *t*-test, two-sided). The box plots display the median by the middle line of the box. The upper boundary of the box indicates the 75% interquartile range, and the lower boundary indicates the 25% interquartile range.

during DR (Fig. 3a, b) while AWA and AWB knockouts still respond to food odor (Supplementary Fig. 4g–j). This result is consistent with AWC neurons perceiving food odor and with the known role of the AWA/B/C neurons in perception of external stimuli[23,38,39].

**DR signaling acts through a pair of enteric neurons**. Previous studies report that serotonin regulates DR and mianserin mediated longevity in liquid culture[17,29]. To further map this pathway that involves the serotonin antagonist mianserin, we first verified that the biogenic amine serotonin is involved in the DR-mediated longevity pathway. We subjected animals lacking *tph-1*, the rate-limiting enzyme necessary to produce serotonin, to DR and mianserin. *tph-1* animals are long-lived compared to wild-type[40] and are not further extended by our DR protocol (Fig. 3c) or mianserin treatment (Supplementary Fig. 5a). These data are supported by the abatement of *fmo-2* induction on DR (Fig. 3d, e) and mianserin (Supplementary Fig. 5b, c) when animals are subjected to *tph-1* (RNAi). As post-mitotic animals, *C. elegans* have a finite number of neurons with discrete connectivity and functions. Three neuronal pairs normally express *tph-1*[41]. The hermaphrodite specific motor neurons (HSN) are located along the ventral tail and regulate egg-laying[42] whereas two head neuron pairs, the amphid neurons with dual sensory endings (ADF) and the neurosecretory motor (NSM) neurons, are involved in modifying behavioral states[26,43,44]. ADF and NSM neurons are also reported to regulate food abundance sensing and food deprivation-mediated longevity[8,9]. To investigate the role of these neuron pairs, we utilized *tph-1* cell-specific knockouts and found that NSM (Supplementary Fig. 5e) but not ADF (Supplementary Fig. 5d) neurons are necessary for DR-mediated longevity. Cell-specific rescue of *tph-1* in a *tph-1* knockout animal suggests that *tph-1* expression in NSM (Fig. 3f), but not ADF (Supplementary Fig. 5f) neurons, is sufficient to promote DR-mediated longevity. Moreover, we find rescuing *tph-1* expression in NSM neurons recapitulates lifespan suppression under food smell (Supplementary Fig. 5h) and blunts its effects when knocked out in the NSM neurons (Supplementary Fig. 5j). However, *tph-1* rescue in ADF neurons does not rescue lifespan suppression by food smell under DR (Supplementary Fig. 5g) but slightly blunts food smell effects when knocked out in ADF (Supplementary Fig. 5i). These results suggest that NSM neurons are the primary serotonergic neurons in modulating *fmo-2* induction and longevity in DR +/− food odor, while ADF neurons may play a smaller role in this pathway as well.

A recent study posits that NSM neurons function similar to enteric neurons with neural projections that directly communicate with the pharynx through a pair of acid-sensing ion channels (ASICs), DEL-3 (WormBase, F26A3.6) and DEL-7 (WormBase, C46A5.2). Signaling through these channels informs the worm to slow locomotion upon contact with food[43]. These data led us to wonder whether the longevity effects of DR also require the ASICs to extend lifespan. We find that *del-7* mutants look phenotypically wild type in their induction of *fmo-2* and lifespan extension, in either DR or DR + food smell (Supplementary Fig. 6a–c). Interestingly, *del-3* mutant worms show abrogated induction of *fmo-2* under DR and did not diminish *fmo-2* induction in response to the smell of food (Fig. 3g–h). These *del-3* mutant animals still exhibit lifespan extension under DR, despite the decreased induction of *fmo-2*, which is not abrogated by the smell of food (Fig. 3i). Together, these data support a model whereby the enteric NSM neurons release serotonin in response to food odor and the lack of this release extends longevity. In addition, the ASIC DEL-3 plays a role in the

NSM to both behaviorally[43] and physiologically respond to food odor and other food signals.

**Dopaminergic CEP neurons are likely involved in odor signaling**. Dopamine synthesis is limited to three neuronal pairs (CEP, ADE, and PDE) in hermaphodites[45]. All three dopaminergic neurons are required for slow movement when encountering a bacteria lawn[46]. To test which dopaminergic neurons respond to food odor, we expressed *cat-2* (WormBase, B0432.5) in CEP, ADE or PDE neurons in the *cat-2* KO strain lacking dopamine production. We find that dopamine synthesis is required for *fmo-2* induction to be suppressed by food odor under DR (Fig. 3j–k) and that rescuing dopamine production using a promotor expressed in CEP neurons and some additional cells[47] restores the suppression of *fmo-2* induction by food odor (Fig. 3j–k). However, *cat-2* rescue in ADE or PDE neurons does not consistently restore the food odor blunting of *fmo-2* induction (Supplementary Fig. 6d–g). These results suggest dopamine produced from CEP neurons is most likely necessary for food odor response.

**Mianserin mimics DR in *fmo-2*-mediated longevity by antagonizing SER-4**. Prior reports suggest that serotonin receptor orthologs *ser-1* (WormBase, F59C12.2) and *ser-4* (WormBase, Y22D7AR.13) are necessary for the lifespan benefits of mianserin in *C. elegans*[48]. We hypothesized that a subset of the serotonin receptor orthologs will also be necessary for mianserin and DR-mediated *fmo-2* induction. After two generations of RNAi treatment, *ser-1* and *ser-4* were the only two receptors that were necessary for *fmo-2* induction on mianserin (Fig. 4a, Supplementary Fig. 7a–c) whereas *ser-4* knockdown most robustly abrogated DR-mediated *fmo-2* induction (Supplementary Fig. 7d, e). Further, we find that *ser-4* RNAi slightly but significantly increases fed lifespan and prevents DR from extending lifespan (Fig. 4b), supporting the hypothesis that mianserin acts as a DR mimetic by antagonizing serotonin signaling that occurs during feeding. We confirmed these knockdown results and find that *ser-4* knockout animals do not respond to the suppression of DR-mediated *fmo-2* induction by food smell (Fig. 4e, f). Next, to investigate whether this effect is mediated by neuronal signaling or intestinal expression, we rescued *ser-4* in *ser-4* knockout animals with tissue-specific promoters and found that only neuronal *unc-119p::ser-4* is sufficient to rescue full induction of *fmo-2* under DR (Fig. 4c, d). This is consistent with serotonergic signaling within the nervous system, and not directly to the intestine, regulating the response to food and food smell.

Single-cell RNA-seq data show *ser-4* is broadly expressed throughout the nervous system[49]. To narrow down the list of potential neurons acting in our pathway, we rescued *ser-4* expression in distinct neuronal populations (promoters used in Supplementary Data 1). We find *ser-4* expressed exclusively in GABAergic neurons is sufficient to rescue food smell suppression of DR-mediated *fmo-2* induction (Fig. 4e, f). Importantly, rescuing *ser-4* expression in neurons producing biogenic amines (Supplementary Fig. 7f, g) or glutamate (Supplementary Fig. 7h, i) does not change *fmo-2* induction compared to the *ser-4* KO. These results suggest one or more GABAergic neurons known to transcribe *ser-4* is responding to serotonin release when food odor is present.

**Thioridazine induces *fmo-2* and extends lifespan through DOP-3/DRD2**. Thioridazine is a compound that antagonizes dopamine receptor D2 (DRD2) in mammals[50–52], and mimics DR by inducing *fmo-2* to increase longevity in nematodes (Fig. 2). Based on its role in mammals, we tested whether nematode DRD2

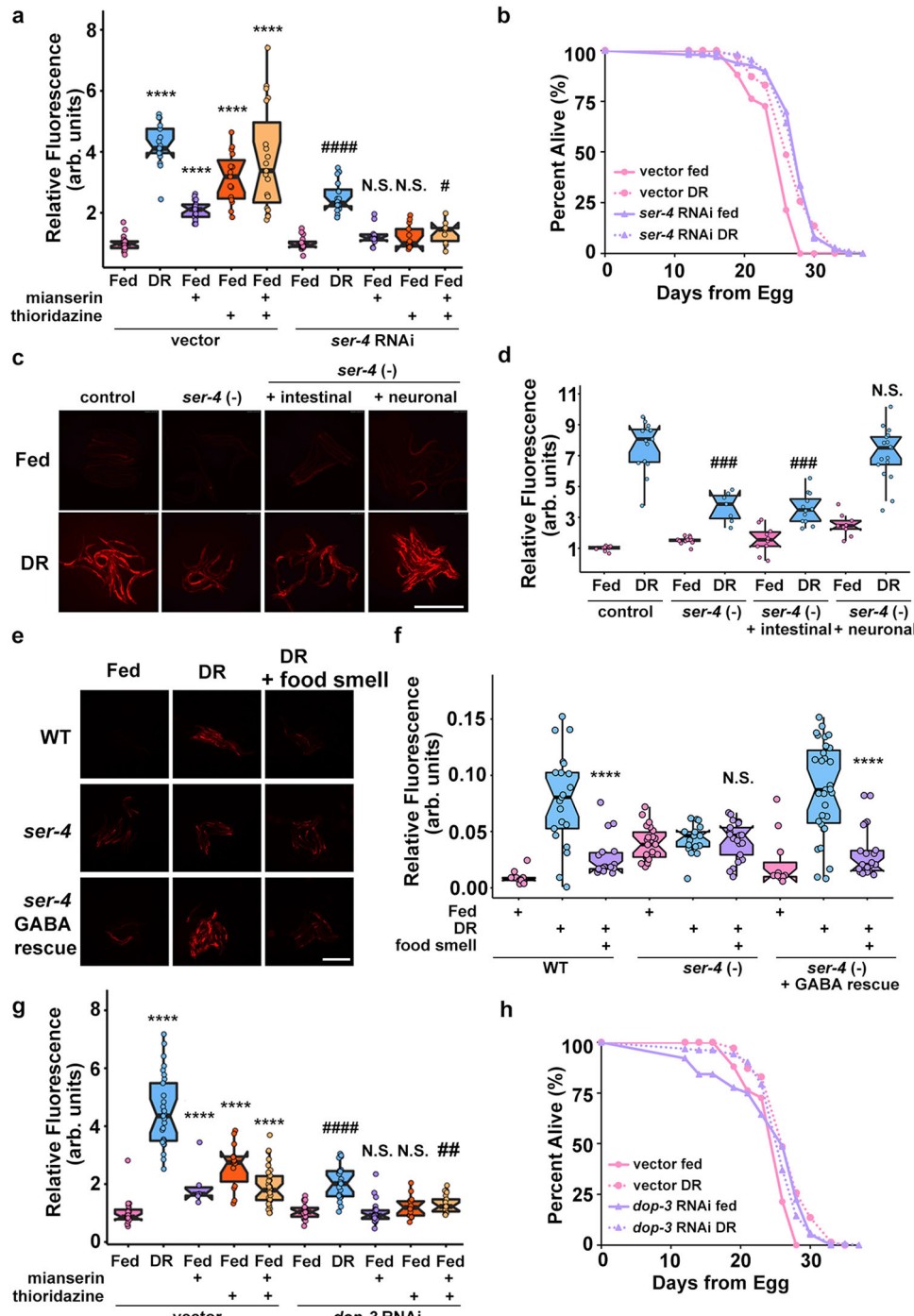

is involved in DR and mianserin-related *fmo-2* induction and longevity. When the DRD2 ortholog *dop-3* (WormBase, T14E8.3) is knocked down by RNAi, *fmo-2* induction is not affected in fed conditions but its induction by DR is diminished, while its induction by thioridazine is completely abrogated (Fig. 4g, Supplementary Fig. 8a). This result is consistent with *dop-3* being required for dopaminergic induction of *fmo-2*. To demonstrate the epistasis of *dop-3* and *ser-4* in the signaling pathway, we combined *ser-4* RNAi with mianserin and thioridazine treatment. The results show that *ser-4* depletion blocks *fmo-2* induction by thioridazine as well as suppresses *fmo-2* induction by mianserin, as expected (Fig. 4a). Similarly, depletion of *dop-3* blocks both mianserin and thioridazine from inducing *fmo-2* (Fig. 4g; Supplementary Fig. 8a). *dop-3* knockouts show similar results to *dop-3* RNAi depletion

(Supplementary Fig. 8a, b). These results support a model where both serotonin and dopamine signaling are epistatic to each other and are each required for full induction of *fmo-2* under DR.

To test whether DOP-3/DRD2 is necessary for lifespan extension by DR and mianserin, we depleted *dop-3* with RNAi under DR and found that *dop-3* depletion increases lifespan but is not further extended by DR (Fig. 4h). Similar to *ser-4* (Fig. 4e–f), we also confirmed that *dop-3* knockout worms do not respond to the suppression of *fmo-2* induction by food odor under DR (Supplementary Fig. 8c, d). Two dopamine receptor D1 (DRD1) orthologs, *dop-1* (WormBase, F15A8.5) and *dop-4* (WormBase, C52B11.3), may also play a role in food odor response (Supplementary Fig. 8c, d). Since DRD1 is not a primary target of thioridazine[50–52], and DRD1 (a Gα$_s$-coupled D1-like receptor

**Fig. 4 5-HT1A receptor *ser-4* and DRD2 receptor *dop-3* act downstream of food odor.** Quantification (**a**) of individual *fmo-2p::mCherry* worms on fed (pink), and DR (blue) treated with 100 µM mianserin (purple), 100 µM thioridazine (orange), or combined (yellow) worms fed vector or *ser-4* RNAi. n = 18 (vector fed), 23 (vector DR), 25 (vector fed + mianserin), 15 (vector fed + thioridazine), 19 (vector fed + mianserin + thioridazine), 16 (*ser-4* RNAi fed), 16 (*ser-4* RNAi DR), 11 (*ser-4* RNAi fed + mianserin), 13 (*ser-4* RNAi fed + thioridazine), 10 (*ser-4* RNAi fed + mianserin + thioridazine) biologically independent animals. p-value = 3.47e-07 (vector fed vs. vector DR), 8.12e-19 (vector fed vs. vector fed + mianserin), 5.91e-08 (vector fed vs. vector fed + thioridazine), 1.30e-06 (vector fed vs. vector fed + mianserin + thioridazine), 5.16e-10 (*ser-4* RNAi fed vs. *ser-4* RNAi DR), 0.04998 (*ser-4* RNAi fed vs. *ser-4* RNAi fed + mianserin), 0.126 (*ser-4* RNAi fed vs. *ser-4* RNAi fed + thioridazine), 0.0165 (*ser-4* RNAi fed vs. *ser-4* RNAi fed + mianserin + thioridazine). Survival curves (**b**) of WT animals on vector RNAi in pink and *ser-4* RNAi in purple on fed (solid lines) or DR (dotted lines) conditions. Images (**c**) and quantification (**d**) of *fmo-2p::mCherry* or *ser-4* +/- tissue-specific rescue on fed (pink) and DR (blue). Scale bar, 1 mm. n = 10 (control fed), 14 (control DR), 14 (*ser-4*(-) fed), 8 (*ser-4*(-) DR), 10 (*ser-4*(-) + intestinal fed), 12 (*ser-4*(-) + intestinal DR), 9 (*ser-4*(-) + neuronal fed), 17 (*ser-4*(-) + neuronal DR) biologically independent animals. p-value = 3.10e-08 (control DR vs. *ser-4* (-) DR), 5.19e-10 (control DR vs. *ser-4* (-) + intestinal DR), 0.131 (control DR vs. *ser-4* (-) + neuronal DR). Images (**e**) and quantification (**f**) of *fmo-2p::mCherry* or *ser-4* KO +/- GABAergic rescue on fed (pink), DR (blue) and DR + food smell (purple). Scale bar, 1 mm. n = 14 (WT fed), 21 (WT DR), 18 (WT DR + food smell), 22 (*ser-4*(-) fed), 17 (*ser-4*(-) DR), 21 (*ser-4*(-) DR + food smell), 11 (*ser-4*(-) + GABA rescue fed), 31 (*ser-4*(-) + GABA rescue DR), 23 (*ser-4*(-) + GABA rescue DR + food smell) biologically independent animals. p-value = 1.72e-05 (WT DR vs. WT DR + food smell), 0.849 (*ser-4*(-) DR vs. *ser-4*(-) DR + food smell), 4.00e-08 (*ser-4*(-) + GABA rescue DR vs. *ser-4*(-) + GABA rescue DR + food smell). Quantification (**g**) of individual *fmo-2p::mCherry* worms on fed (pink), and DR (blue) treated with 100 µM mianserin (purple), 100 µM thioridazine (orange), or combined (yellow) worms fed vector or *dop-3* RNAi. n = 23 (vector fed), 29 (vector DR), 6 (vector fed + mianserin), 14 (vector fed + thioridazine), 34 (vector fed + mianserin + thioridazine), 24 (*dop-3* RNAi fed), 23 (*dop-3* RNAi DR), 26 (*dop-3* RNAi fed + mianserin), 15 (*dop-3* RNAi fed + thioridazine), 21 (*dop-3* RNAi fed + mianserin + thioridazine) biologically independent animals. p-value = 1.69e-16 (vector fed vs. vector DR), 0.0268 (vector fed vs. vector fed + mianserin), 8.34e-07 (vector fed vs. vector fed + thioridazine), 1.03e-07 (vector fed vs. vector fed + mianserin + thioridazine), 1.88e-08 (*dop-3* RNAi fed vs. *dop-3* RNAi DR), 0.542 (*dop-3* RNAi fed vs. *dop-3* RNAi fed + mianserin), 0.166 (*dop-3* RNAi fed vs. *dop-3* RNAi fed + thioridazine), 0.00648 (*dop-3* RNAi fed vs. *dop-3* RNAi fed + mianserin + thioridazine). Survival curves (**h**) of WT animals on vector RNAi in pink and *dop-3* RNAi in purple on fed (solid lines) or DR (dotted lines) conditions. ****$P < .0001$ (Welch Two Sample *t*-test, two-sided). ###$P < 0.001$ when compared to *ser-4* (-) or *ser-4* (-) + intestinal fed (Welch Two Sample *t*-test, two-sided); #$P < 0.05$, ####$P < 0.0001$ when compared to *ser-4/dop-3* RNAi fed (Welch Two Sample *t*-test, two-sided). The box plots display the median by the middle line of the box. The upper boundary of the box indicates the 75% interquartile range, and the lower boundary indicates the 25% interquartile range.

inducing cAMP[53,54]) has antagonistic effects of DRD2 (a Gα$_{i/o}$-coupled D2-like receptor inhibiting cAMP[53,54]) on behaviors in mammals[55] and *C. elegans*[56], *dop-1* and *dop-4* might be acting in a parallel pathway in food perception. Together, these results suggest that dopamine and serotonin signaling interactively suppress *fmo-2* expression to limit lifespan when food and/or odor are present.

To further test the target specificity of mianserin and thioridazine in *C. elegans*, we measured the induction of *fmo-2* by mianserin or thioridazine in *ser-4* or *dop-3* knockout worms (Supplementary Fig. 8E, F). The results show that *fmo-2* induction by mianserin or thioridazine is decreased in *ser-4* or *dop-3* knockout worms. This further suggests that induction of *fmo-2* by mianserin and thioridazine requires both *ser-4* and *dop-3*.

In addition to *ser-4*, the octopamine receptor *ser-3* (WormBase, K02F2.6) is also antagonized by mianserin and is required for lifespan extension by mianserin[29]. Thus we asked whether *ser-3* is involved in this pathway by testing whether mianserin or thioridazine induces *fmo-2* through *ser-3*. The results show that *fmo-2* induction by mianserin or thioridazine is decreased but not blocked in *ser-3* knockouts compared to wild type worms (Supplementary Fig. 8e, f). This is consistent with *ser-3* playing a role in *fmo-2* induction by mianserin and thioridazine. We interpret this as *ser-3* acting downstream of both drugs but not as the primary/only downstream receptor. This result agrees with previous reports that suggest octopamine signaling is downstream of dopamine in the food availability response[57,58] and interacts with serotonin in regulating body fat[59] and aversive behaviors[60]. Based on this, we tested whether octopamine signaling is involved in sensing food odor. We found that the octopamine synthesizing enzyme *tbh-1* (WormBase, H13N06.6) and to a lesser extent *ser-3* knockout animals lack suppression of DR-mediated *fmo-2* induction in the presence of food odor. This suggests that octopamine is involved in this pathway but plausibly through multiple octopamine receptors (Supplementary Fig. 8g, h).

**Mianserin extends *D. melanogaster* lifespan.** Having identified serotonin and dopamine antagonism upstream of *fmo-2*

induction under DR, we were curious whether these relationships might be conserved. Similar to data in worms, recent data in the vinegar fly *D. melanogaster* show that altered serotonin signaling can change the ability to assess caloric quality and modulate lifespan[27]. As we found a narrow range of effective doses in worms (Fig. 2f), we tested a higher dose of mianserin in vinegar flies (2 mM) for its effect on Fmo2 induction. The resulting data show that both mianserin and fasting (DR) increase expression of fly *fmo-2* (NCBI Gene 35561) expression (Fig. 5a), but not *fmo-1* (NCBI Gene 37814) (Supplementary Fig. 9a). We then asked whether mianserin could also extend lifespan in flies. Using several concentrations, we find a positive correlation between mianserin dosage and increased lifespan until reaching a detrimental level of serotonin antagonism (Fig. 5b, Supplementary Fig. 9b–d). We also find a comparable dose response among male and female flies. We note that mianserin treatment does not significantly alter food consumption (Supplementary Fig. 9e, f), as measured by the Fly Liquid-Food Interaction Counter (FLIC) assay[61]. Together, these results are consistent with conserved induction of *fmos* by mianserin and DR, in addition to conserved lifespan extension.

## Discussion

Our experimental data in *C. elegans* support a model where dietary restriction (sDR) leads to decreased serotonin release from the enteric NSM neurons and decreased dopamine release, likely from the CEP neurons. These decreases lead to reduced serotonin binding to the SER-4/5-HT1A receptor and a reduction in dopamine signaling to downstream DOP-3/DRD2 receptors. Subsequently, the loss of binding to SER-4 and DOP-3 causes a downstream induction of *fmo-2* and extension of lifespan.

It is notable that both SER-4 and DOP-3 receptors are known to dampen adenylyl cyclase activity when bound, thus the lack of signal will increase the probability of excitement of the cell expressing these receptors. In our working model, we hypothesize that the presence of food odor acts as a signal sensed by AWC neurons, which signal upstream, downstream, or in parallel with

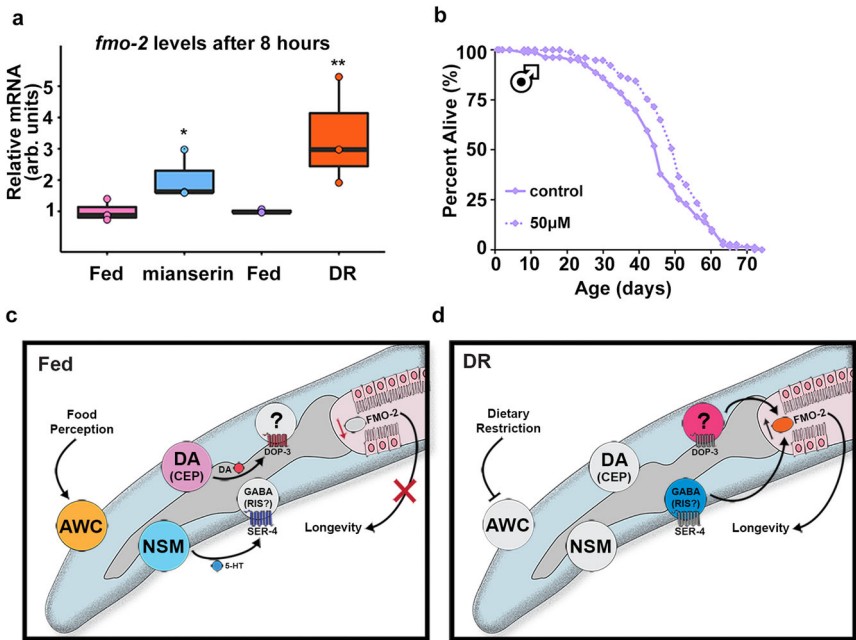

**Fig. 5 Serotonin antagonist mianserin induces FMO and improves health in Drosophila.** Fmo-2 mRNA levels (**a**) after eight hours of 2 mM mianserin (blue) or starvation (orange) compared to water controls (pink and purple, respectively). $n = 3$ (fed, mianserin, fed, DR) biologically independent experiments. $p$-value = 0.0365 (fed vs. mianserin), 0.00914 (fed vs. DR). Survival curves of male flies treated with water (solid line) or 50 μM (dotted line) mianserin (**b**). Panels **c** and **d** depict the current model for the cell non-autonomous DR/food odor signaling pathway. "DA" stands for dopaminergic neuron. *$P < .05$, **$P < .01$ when compared to control or fed (Welch Two Sample $t$-test, two-sided). The box plots display the median by the middle line of the box. The upper boundary of the box indicates the 75% interquartile range, and the lower boundary indicates the 25% interquartile range.

NSM and CEP neurons that release serotonin and dopamine, respectively (Fig. 5c). The released serotonin and dopamine bind to the serotonin receptor SER-4 on one or more GABAergic neurons and the dopamine receptor DOP-3 respectively to suppress *fmo-2* induction and longevity (Fig. 5c). We note that the GABAergic neuron, RIS, expresses more *ser-4* than any other neuron and thus is a good candidate for a role in this signaling pathway[62]. In contrast, under sDR, food signals are likely decreased to a level insufficient to excite the neural circuits involving AWC, NSM, and CEP neurons. Decreased serotonin and dopamine release under DR can be mimicked by serotonin and dopamine antagonists that induce *fmo-2* and promote longevity (Fig. 5d). Critically, these data highlight that understanding how the nervous system evaluates and appropriately integrates large amounts of external stimuli, like the availability of food, allows us to target the sensory-integration processes to mimic key aspects of pro-longevity pathways.

A previous report finds that mianserin antagonizes octopamine and serotonin to increase lifespan and that its mechanism overlaps with DR[29]. Our results reinforce this previous finding and further identify key neurons, neurotransmitters and receptors that respond to food availability. We also demonstrate FMO-2, a monooxygenase that is sufficient to extend lifespan in worms[7], as a converging downstream effector of serotonin and dopamine signaling for longevity benefits.

Our findings suggest that NSM is important for both DR and food odor effects on lifespans and ADF may play a smaller role as well (Fig. 3f; Supplementary Fig. 5d–j). It is notable that a recent report shows that ADF neuron activity is responsible for food smell effects on longevity[9]. This discrepancy may be due to the difference in methodology. We tested the role of serotonin produced by NSM or ADF by both adding the serotonin synthetase TPH-1 (*tph-1* cell-specific rescue) and subtracting TPH-1 (*tph-1* cell-specific knockout) in these neurons to specifically target

serotonin production in NSM or ADF neurons. In comparison, the other study ablated the output of serotonergic neurons using tetanus toxin. Tetanus toxin is a broad signaling inhibitor that will likely affect all signals from serotonergic neurons. It is also interesting to note that both the NSM neurons and the NSM enriched acid-sensing ion channels (ASICs) DEL-3 are involved in both sensing food odor for longevity (Fig. 3) and food ingestion in behavioral changes as previously reported[43]. DEL-3 and DEL-7 are required for NSM activation of post-food encounter slowing, however may not be involved in the behavioral changes from food odor[43].

It is intriguing that dopamine and serotonin signaling interactively induce *fmo-2* and extend lifespan in a common dietary restriction-mediated pathway. In nematodes, slowing locomotion in the presence of food is thought to be distinctly regulated by mechanosensation leading to dopamine release while dwelling behavior is potentiated by serotonin[46]. Significant scientific effort has identified much of the circuitry these neurotransmitters use to promote changes in chemotaxis and egg-laying[26,43,63–65]. Those data suggest worms can interpret and implement a diverse set of responses to their changing environment. In mammals, SER-4/5-HT1A receptor activation increases dopamine release throughout the brain[66,67]. Similarly, recent work shows release of serotonin and dopamine in the human brain influence non-reward-based aspects of cognition and behaviors like decision making[68]. These findings support a conserved link between these two neurotransmitters in regulating complex phenotypes like aging.

Interestingly, dopamine may have a more complex role in aging. Knockout of dopamine reuptake receptor *dat-1*, which leads to increased dopamine signaling, and knockout of *cat-2* that decreases dopamine synthesis, both can shorten lifespans in worms[69]. Dopamine antagonists are found to extend worms lifespan in a pharmacological screen[17]. Our data also show that

dopamine antagonists can promote longevity by inducing *fmo-2* (Fig. 2). It is possible that dopamine has both positive and negative effects on lifespan depending on the context.

We note that although antagonists of dopamine and serotonin signaling induce *fmo-2* in worms, decreasing this signaling by *tph-1*, *ser-4*, *dop-3* or *del-3* knock down or even AWC ablation is not sufficient to induce *fmo-2* under fed conditions (Figs. 3d, e, g, h; 4a, g, Supplementary Fig. 10). While not directly tested here, we hypothesize that two potentially overlapping possibilities could explain this: 1) Because there are multiple sensory inputs for food perception, loss of just the food smell pathway is not sufficient to produce a meaningful output, and 2) when cells or genes are absent or highly knocked down throughout development, the signaling networks they regulate undergo rewiring to respond to cues through different pathways. These hypotheses could both play some role in our observed results, and we plan future research to test them.

It is also intriguing that one of these drugs, mianserin, induces Fmo genes in flies. This leads to increased lifespan. Since mianserin treatment extends fly lifespan, we suspect it acts through a similar mechanism, serotonin antagonism, to mimic DR. This hypothesis is bolstered by *fmo-2* induction under acute mianserin exposure and fasting, analogous to what we see in *C. elegans*. We note that in combination with DR, mianserin does not increase longevity of worms any further. It is not known whether FMOs or 5-HT1A receptors are necessary for mianserin or DR-mediated longevity in flies, but 5-HT2A receptors are necessary for proper food valuation[27], suggesting that altering serotonin signaling may prove fruitful in future studies.

Mammals and *C. elegans* share a single common ancestral Fmo[21] and mammalian Fmos share similar homology to *C. elegans fmo-2*, with Fmo5 (NCBI Gene 14263) having the highest % identity. It will be interesting to investigate whether mianserin is beneficial for health and longevity in mammals. To achieve this goal, it is imperative that we understand the causative changes of pro-longevity drugs, such as atypical serotonin antagonists, that are known to have pleiotropic effects in humans. In addition to providing the potential for long-term health benefits, this knowledge will benefit our understanding of serotonin and dopamine signaling networks that affect many human processes and diseases outside of aging.

## Methods

**Strains and growth conditions**. Standard procedures for *C. elegans* strain maintenance[70] were used. In detail, strains were kept at 20 °C in a temperature-controlled incubator and were fed on Escherichia coli (OP50) seeded on solid nematode growth medium (NGM). Worms were picked or transferred gently by a platinum wire. Additionally, worms were exposed to the smell of OP50 or HB101 when indicated in the experiments. Supplementary Data 1 includes a list of the strains and RNAi conditions used in this study. All genotypes were confirmed using PCR.

**fmo-2p::mCherry construct**. We PCR amplified *mCherry* from pHG8 and the *fmo-2* promoter from the worm gDNA under the *fmo-2* promoter and cloned them into pdonr221 and P4-P1r, respectively. From here, they were combined using Gateway LR cloning (Invitrogen) to create *fmo-2p::mCherry*::unc-54 3'UTR on PCFJ150.

**AWA/B/C ablation constructs**. We purchased donor plasmid P*mec-18*::caspase-3 (p12)::nz [TU#813] from Addgene (Plasmid #16082) and P*mec-18* cz::caspase-3 (p17) [TU#814] from Addgene (Plasmid #16083), and used Gibson cloning (NEB) to replace P*mec-18* with P*odr-7*. Three constructs P*odr-7*::caspase-3(p12)::nz, P*odr-7*::cz::caspase-3(p17), and P*odr-7*::GFP were co-injected with fluorescent co-injection marker *myo-3p*::GFP (20 ng/μL) into the *fmo-2p::mCherry* transcriptional reporter strain to generate AWA genetic ablation strains. Similarly, P*str-1* was constructed into TU#813 and TU#814 to replace P*mec-18*. Three constructs P*str-1*::caspase-3(p12)::nz, P*str-1*::cz::caspase-3(p17), and P*str-1*::GFP were co-injected with fluorescent co-injection marker *myo-3p*::GFP (20 ng/μL) into the *fmo-2p::mCherry* transcriptional reporter strain to generate AWB genetic ablation strains. The AWC genetic ablation strain (oyIs85 [*ceh-36*p::TU#813 + *ceh-36*p::TU#814 + *srtx-1*p::GFP + *unc-122*p::DsRed]) is from CGC, PY7502. AWC

genetic ablation strain was crossed with *fmo-2p::mCherry*. All plasmids were verified via restriction digest and sanger sequencing. ApE files available upon request.

**SER-4 and CAT-2 rescue constructs**. We purchased donor plasmid pPD117.01 from Addgene and used Gibson cloning (NEB) to swap out promoters driving cDNA of SER-4::SL2::GFP (on backbone) expression. We used the *unc-119* promoter to target all neurons and the *vha-6* promoter to target the intestine. We used *unc-47*, *vglu-2*, and *cat-1* promoters to target SER-4 expression in GABAergic, glutamatergic, and biogenic amine neurons, respectively. Similarly, we constructed promoters driving cDNA of CAT-2::SL2::GFP expression. We used *swip-10*, *trpa-1* and *tax-2* promoters to target CAT-2 expression in CEP, ADE, and PDE neurons, respectively. We used *cat-1* promoter to rescue CAT-2 expression in all three dopaminergic neurons (CEP, ADE, and PDE). All plasmids were verified via restriction digest and sanger sequencing. ApE files available upon request.

**Microinjection**. Single-copy integration of *fmo-2p::mCherry*::unc-54 3'UTR on PCFJ150 using the ttTi5605 (EG6699) Mos allele was performed as previously described[71]. Overexpression transgenic animals were generated by injecting PureLink (Invitrogen) miniprepped DNA clones (~50 ng/μL) with fluorescent co-injection marker *myo-2p*::mNeonGreen (15 ng/μL) or *sur-5p*::sur-5::NLSGFP (20 ng/μL) and junk DNA (up to 100 ng/μL) into gonads of day 1 gravid adult hermaphrodites. Standard protocols were followed to isolate and obtain stable over-expression mutants[72]. Because transgene expression can vary substantially, we typically characterized 2–4 independent transgenic lines per experiment.

**Lifespan measurements**. Lifespans were carried out as previously described with minor modifications[73]. Briefly, 20–30 gravid adult animals were placed on NGM plates for a timed egg-lay. After 12–16 h, these animals were removed. Once their progeny reached late L4/early adult stage, animals were transferred to plates with 33 μL of 150 mM fluorodeoxyuridine (FUdR) and 100 μL of 50 mg/mL Ampicillin per 100 mL NGM to prevent the development of progeny and growth of bacteria. Roughly 75 worms were placed on each NGM + FUdR plate seeded with concentrated bacteria (×5). A minimum of two plates per strain per condition were used per replicate. Lifespan plates were transferred periodically during early adulthood to prevent starvation and avoid contamination. Animals were scored as dead and removed from the experiment when they did not move in response to prodding under a dissection microscope. It is notable that FUdR can extend lifespan and is a confounding factor contributing to lifespan extension[74], however, to avoid matricide under DR, we include FUdR in the lifespan plates in this study. This factor should be considered when interpreting the results.

**RNAi knockdown**. The RNAi feeding bacteria were obtained from the Ahringer *C. elegans* RNAi feeding library. All RNAi plasmids were sequenced to verify the correct target sequence. Animals were exposed to RNAi plates from egg on plates consisting of NGM supplemented with 1 mM β-D-isothiogalactopyranoside (IPTG) and 25 μg/ml carbenicillin. At late L4 stage of development the animals were transferred to plates containing freshly seeded RNAi bacteria plus FUdR. RNAi efficiency of genes expressed in neurons is not validated by qPCR but RNAi phenotypes are recapitulated by gene knockouts. It is notable that RNAi feeding bacteria are HT115 and are reported to modify lifespan compared to OP50[75,76]. This should be taken into consideration when comparing the lifespan data of RNAi feeding for gene knock down with the lifespan data of gene mutants fed on OP50.

**PFA treatment**. In order to metabolically kill OP50 in food smell assays, bacteria cultures were treated with 0.5% PFA. After 16 h of shaking, 50 mL of the bacteria were aliquoted into 250 mL Erlenmeyer flasks. 32% PFA was added to the flasks to get the desired final PFA concentration (e.g., 781 μL of PFA was added to get the final concentration of 0.5% PFA). PFA-treated bacteria were shaken at 37 °C for 1 h and then transferred to 50 mL conical tubes, centrifuged and washed with LB five times to remove residual PFA before seeding.

**Drug treatments**. Recent reports show improved health outcomes and longevity in nematodes treated with mianserin[48], but only in liquid culture[32]. As our studies are on agar plates, we modified previous protocols by adding mianserin, thioridazine or trifluoperazine before pouring NGM agar plates. Without proper dosing, these neurotransmitter antagonists can cause off-target effects like fleeing, especially when combined with DR. All subsequent *C. elegans* experiments were performed at 50 μM of mianserin and 25 μM of thioridazine unless otherwise noted. All drugs were purchased from Sigma-Aldrich and were initially dissolved in milliQ water at 2 mM (mianserin) or 100 mM concentration (DRD2 antagonists), aliquoted, and stored at −20 °C.

**Dietary restriction (DR) lifespan treatments**. Lifespan DR assays were performed like other lifespans until day two of adulthood when the worms were transferred to plates with $10^9$ cfu/ml seeded lawns and transferred every other day four times. This form of DR is termed solid DR (sDR)[77]. For short-term DR assays, worms were starved for 8 (real-time PCR) or 20 h (slide microscopy). We added

100 µL of 10 mM palmitic acid (Sigma-Aldrich) dissolved in 100% EtOH to the outer rim of the plate to prevent fleeing.

**Attractant, repellant, and neutral smell treatments.** Fed and DR plates were prepared using NGM plates with palmitic acid. Odorants were chosen from previously published work isolating secreted compounds from the *E. coli* strain HB101[23,24]. All concentrations of attractant, repellant, and neutral chemicals were dissolved in 100% ethanol (more details in Supplementary Data 3). A small pad of NGM agar (2 mL) was poured on the lid of each plate and allowed to solidify before 100 µL of each smell concentration was added to the agar pads. Plates were prepared the day prior to use to allow the ethanol solutions to dry. Young adult *fmo-2p::mCherry* worms were placed on fed and DR plates and exposed to each smell for 20 h before fluorescent microscopy images were taken.

**Slide microscopy.** All images in this study were acquired using Leica Application Suite X software and Leica scope with >15 worms/treatment at x6.3 magnification. Worms were paralyzed in 0.5 M sodium azide (NaN3). Fluorescence mean comparisons were quantified in ImageJ bundled with 64-bit Java 1.8.0 using the polygon tool and saved as macros. Data were plotted by R version 4.1.0, Microsoft Excel 365, Adobe Photoshop 2021, and Adobe Illustrator 2021.

**Real-time PCR.** 500 N2 worms per biological replicate were transferred at young adulthood, 2.5 days post-hatch, to FuDR plates either seeded with OP50 of fed or DR condition or poured with the addition of 50 µM mianserin or thioridazine. Alternatively, N2 worms were exposed to RNAi plates from egg on plates consisting of NGM supplemented with 1 mM β-D-isothiogalactopyranoside (IPTG) and 25 µg/ml carbenicillin for two generations. Worms were harvested in 50 µL of M9 and flash frozen in liquid nitrogen after 8 hours of exposure. Samples were freeze-thawed three times in Trizol reagent (Invitrogen) and RNA was extracted following standard phenol-chloroform protocols from the manufacturer. Superscript reverse transcriptase II (Invitrogen) was used to synthesize cDNA. 600 ng of cDNA/sample were used with PowerUp SYBR Green Master Mix (Applied Biosystems) was used in the quantitation with fmo-2 forward primer ACGAAACGA ATGAGTCGTCAGT and reverse primer AGAGCAGACAGAACGCCAT.

Canton-S flies were mated and reared on standard food for 2 weeks before separating the flies by sex onto SY10 food with 20 flies/vial. Flies were acclimated to the vials for 24 h before being transferred to SY10 vials coated with 2 mM mianserin or water (control) or vials containing 2% agar to mimic dietary restriction. After 8 h on these treatments, flies were frozen at −80 °C overnight. Fly heads and bodies were then separated by vortexing and dissection by forceps (all samples and materials were kept on dry ice throughout). Each treatment contained 3 biological replicates composed of 10 bodies each. Trizol Reagent (Invitrogen) was used in the RNA extraction, the MultiScribe Reverse Transcriptase kit (Applied Biosystems) was used to synthesize the cDNA, and the real-time PCR analysis used PowerUp SYBR Green Master Mix (Applied Biosystems) and a StepOne Plus Real-time PCR system (Applied Biosystems) primers of *fmo-1* (forward primer GCGA TAGGATGGGCAAACTG and reverse primer CCCGGAAGTGGAGCAAATTC) and *fmo-2* (forward primer CGCAACCAGAAGAAAGCACA and reverse primer TGCTCCTGTACGTGTCCAAT).

**Fly husbandry.** The laboratory stock Canton-S was used in the lifespan and molecular experiments. Flies were maintained on standard food and housed at 25 °C and 60% relative humidity in a 12:12 h light-dark cycle.

**Fly survival assays.** For lifespan measurements, flies were reared under controlled larval density and collected onto standard food within 24 h of eclosion. Flies were mated for 2–3 days then sorted by sex under light $CO_2$ onto vials containing standard food used in lifespan experiments (10% sucrose/10% yeast, or SY10), according to well-establish lifespan protocols[78]. Flies were transferred to fresh food every 2–3 days. At the beginning of the lifespan, mianserin was dissolved in water at a 1 mM stock concentration and stored at −20 °C. Weekly aliquots were prepared and diluted with water to yield the final concentrations of 20–80 µM. 100 µL of the drug solution (or water for the control) was added to the top of each vial and kept at room temperature to dry for approximately 2 h before transferring the flies.

**Statistical analyses.** All box plots show individual data points while the box represents SEM (centered on the mean), and whiskers represent 10%/90%. Comparisons between more than two groups were done using ANOVA. For multiple comparisons, Welch's Two Sample *t*-test (two-sided) was used, and p values are $*p < 0.05$, $**p < 0.01$, $***p < 0.001$, and $****p < 0.0001$. For lifespan assays, the statistical groupwise and pairwise comparisons among survivorship curves were performed by Online application for survival analysis (OASIS 2)[79]. P values were obtained using the log-rank analysis (select pairwise comparisons and group comparisons or interaction studies) as noted. Summary lifespan data, sample size (*n*), and statistics are included in Supplementary Data 2. Measurements were taken from distinct samples. The box plots display the median by the middle line of the box. The upper boundary of the box indicates the 75% interquartile range, and the lower boundary indicates the 25% interquartile range.

## Data availability

All data generated or analyzed during this study are included in this published article (and its Supplementary Information files). Source data are provided with this paper.

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

## Acknowledgements

We thank Dr. Cori Bargmann for providing the tph-1 cell specific knockout strains in ADF or NSM neurons. We thank Dr. Kaveh Ashrafi for providing pKA805[srh-142p::TPH-1] and pKA807[ceh-2p::TPH-1] constructs for injections. We thank Suny-Biotech for generating some strains by injections for this study. We thank all members of the Leiser laboratory for suggestions and discussions. Research reported in this publication was supported by NIA R01 AG059583 and the Paul F. Glenn Foundation for Aging Research to S.F.L., and NIA F31060663 and the Rackham Predoctoral Fellowship from the University of Michigan to H.A.M.

## Author contributions

H.A.M., S.H., and S.F.L. developed the conceptual framework and wrote the manuscript. H.A.M., S.H., E.S.D., M.L.S., A.M.T., A.S.M., S.B., and S.F.L. contributed to data collection and analysis. S.D.P contributed to the data interpretation and discussion. H.A.M., S.H., and E.S.D. prepared the figures and tables. All authors reviewed and approved the manuscript.

## Competing interests

The authors declare no competing interests.
