## [Peer Review File · Nature Communications]

REVIEWER COMMENTS

Reviewer #1 (Remarks to the Author):

General Comments:

Miller et al uses *C. elegans* as a model system to study serotonergic and dopaminergic signaling in DR and life span. They propose a model in which DR antagonizes biogenic amine signaling to upregulate *fmo-2* and increase life span. The reviewer felt that the manuscript fails to establish a definitive mechanism and has built on extant understanding, making incremental progress, may not be sufficient to be considered at this stage. The manuscript may benefit for proper structuring and increased clarity.

There are some aspects that are already known in the field. Some examples are:

A) While the authors were surprised to find food smell suppressing DR life span, this has already been documented (Libert et al., 2007 doi: 10.1126/science.1136610; Smith et al., 2008, doi: 10.1186/1471-213X-8-49, Zhang et al., 2021; doi: 10.1038/s43587-021-00039-1; Park et al., 2021, doi: 10.1016/j.isci.2020.101979). It is important that these are referenced.

B) Mianserin, that blocks neural signalling by the neurotransmitter serotonin, is known to increase life span in a manner similar to DR (Petrascheck et al., 2007, doi: 10.1038/nature05991)

Specific comments pertaining to the experiments, hypothesis and interpretations:

- 1) Why metabolically active bacteria are required to produce these smells? How can one show that metabolically inactive bacteria do not emit the compounds that worms smell? Does the solid plate DR indeed reduce smell of bacteria or the levels of attractants?
- 2) It appears that the average life span of wild type worms is in the range of 25 days where as *C. elegans* typically have an average life span of less than 20 days. Is there a particular reason for this?
- 3) The authors mention that *fmo-2* is sufficient for life span extension. So, do compounds that induce *fmo-2* expression, like neutral and repellent compounds, increase life span?
- 4) The study is based on the hypothesis that DR decreases biogenic amine signaling leading to higher *fmo-2* and life span. Is it not possible that DR increases *fmo-2* independently while biogenic amine negatively regulates *fmo-2*? Food smell may activate biogenic amine signaling that negatively regulates *fmo-2*. If moderate activation of *fmo-2* is sufficient for life span extension, one may not see synergy between DR and the antagonist in this model.
- 5) In line 157, the authors hypothesize that DR reduces food smell. However, there is no evidence in support of this hypothesis. Can this be experimentally validated by measuring the attractant levels on DR plates?
- 6) Was RNAi efficiency checked by quantitative PCR under each condition?
- 7) Why is *fmo-2* not induced on *tph-1* knock down when the hypothesis is that DR induces *fmo-2* by reducing biogenic amine? Similar question arises for *del-3(-)*.
- 8) The legends of S5 D and E are probably swapped.
- 9) The conclusion of line 186 about food smell is probably not correct as no experiment with food smell was conducted under these conditions.
- 10) Why is *fmo-2* and life span decoupled in case of *del-3(-)*?
- 11) The authors mention that mianserin mimics DR. Are all aspects of DR mimicked by mianserin? They need to check the metabolic reprogramming aspect of DR before making this comment. It is possible that biogenic amine signaling is an independent signaling that partially overlaps with how DR affects life span.
- 12) The authors use pan-neuronal rescue of *ser-4* for their experiment. It will be more informative to identify the specific neurons.
- 13) While the premises on which this study is based on is tissue-to-tissue communication of food signal. The experiment with mammalian cell culture makes no sense. It raises the question why it is cell non-autonomous in worms and flies.
- 14) The authors have used *fmo-2* tagged transgenic line for all experiments. They should back their observation up with quantitation of the *fmo-2* transcript abundance.
- 15) Why the values of y-axis in Figure 1 for fluorescence measurement so varied?
- 16) Figure S2B: induction of *fmo-2* is more in DR compared to fed conditions. Then, why the mean fluorescence of *fmo-2* of the controls is more in fed condition while reduced in

DR?

17) Some statistical comparisons are missing. Eg. Fig 2B and C. between DR+smell and DR+smell+antagonist.

18) A recent study, Zhang et al 2021 shows the involvement of ADF neurons in food odour-dependent suppression of DR life span. The authors need to discuss why their results are different from the published one. Have the authors checked the involvement of this neuron in DR+food smell condition?

19) Figure 4B: ser-4 receptors are required in neurons to exhibit DR mediated fmo-2 induction. Have you checked whether ser-4 receptors are also involved in fmo-2 modulation under DR+smell scenario?

20) Figure 4(E,F): shows that dopaminergic signalling is required for induction of fmo-2 under DR. Have you checked which of the CEP, ADE or PDE dopamine synthesizing neurons is involved in smell perception during DR?

Reviewer #2 (Remarks to the Author):

This manuscript builds on prior work on the role of serotonin and dopamine in *C. elegans* aging in the context of dietary restriction. The principal areas of novelty lie in connecting dietary restriction and sensing of bacterial odors to the regulation of fmo-2, a flavin-containing monooxygenase, as well as in evidence that aspects of flavin-containing monooxygenase regulation are conserved in other species.

Major issues:

1. Mianserin and thioridazine induce fmo-2 expression at 100µM (Figure 2A-E and 4A) but at this concentration these drugs do not have an effect on lifespan (Fig. 2F-G). Thus, this data does not support the conclusion that the induction of fmo-2 expression by these drugs leads to lifespan extension. This issue should be addressed by examining fmo-2 expression under concentrations of mianserin and thioridazine that extend lifespan (e.g., 25µM).

2. The roles of serotonin in dietary restriction and aging, and of dopamine in aging have been previously described. This paper has overlooked precedents in the literature that should be described:

a. tph-1 has been previously shown to have a role in lifespan changes due to dietary restriction. Zhang et al., Nature Aging 2021 (<https://doi.org/10.1038/s43587-021-00039-1>) showed in that tph-1 mutants under DR do not shorten lifespan in response to bacteria odor, using a similar set up as this manuscript. Entchev et al., eLife 2015 (<https://doi.org/10.7554/eLife.06259>), showed that tph-1 mutants affect lifespan under DR and that tph-1 in NSM plays is important in this process.

b. Multiple drugs targeting HTR2A receptors (including mianserin) and multiple drugs targeting DRD2 (including thioridazine) have previously been shown to extend *C. elegans* lifespan in Ye et al., Aging Cell 2014 (<https://doi.org/10.1111/accel.12163>).

It is gratifying that the authors could reproduce results from the literature, but such prior work should be described to provide a more accurate context for the results presented.

3. Benedetto et al., PLOS Genetics 2010 (<https://doi.org/10.1371/journal.pgen.1001084>) showed that loss of dopamine signalling via dat-1 and cat-2 mutations shorten lifespan, which differs from the lifespan extensions due to thioridazine treatment. Together, these apparent discrepancies suggest a more complex role for dopamine which should be discussed.

4. In contrast to Rhoades et al., (ref 30 in the manuscript), which describes NSM as a neuron that is activated by food ingestion, the results presented in this manuscript suggests that NSM is also important for sensing food odors (or relaying that information). Moreover daf-6 mutants, where amphid sensory neuron function are disrupted, maintain the ability to sense food odors (presumably by NSM). This difference

in the role of NSM (sensing odor versus ingestion) should be discussed. The manuscript uses the term "food perception" – this is vague and should be replaced with the type of perception (food ingestion, odor etc.) to be more precise.

5. The supplementary tables were not included in the files made available to me. Thus, it was not possible to assess the statistical data such as the number of trials, no. of animals tested etc. Similarly, there was no information on the strains used in this paper, including alleles, transgenes, and number of outcrosses.

Minor issues:

The paper draws conclusions by combining results from both RNAi feeding and mutant data. Results from these different types of experiments are difficult to compare because RNAi involves feeding animals a different type of bacteria (HT115) with known impact on lifespan (Maier et al., PLOS Biology 2010 <https://doi.org/10.1371/journal.pbio.1000376>; Pang & Curran, Cell Met. 2014 <https://doi.org/10.1016/j.cmet.2013.12.005>). The authors should acknowledge this technical issue as a caveat in the methods section.

FUDR is a confounding factor in lifespan studies (Anderson et al, Mech Age Dev 2016 <https://doi.org/10.1016/j.mad.2016.01.004>). This should also be acknowledged as a caveat in the methods.

It is surprising that RNAi of neuronal genes is so effective in this study, since neurons are resistant to RNAi in *C. elegans*. Can the authors briefly explain how they achieved effective knockdown (Supplementary Table 3 was not available for me, perhaps this was addressed there), or whether a slight reduction is sufficient?

The text implies that *unc-13* is required only for synaptic vesicle exocytosis, which is not accurate. In fact, *unc-13* is also required for dense core vesicle exocytosis as well (Li & Kim, Wormbook 2008 <https://doi.org/10.1895/wormbook.1.142.1>)

The discussion describes an on-off toggle state for food perception. This is not supported by the experiments. Since only 2 conditions were tested, it is not possible to know if additional intermediate responses are possible. However, many papers in the field have shown graded responses in lifespan extension under multiple levels of DR, which is not compatible with a toggle model.

Line 246: Not clear what dysregulation means in this context. In Figure S7F-G, S8B-D, *fmo-2* expression does not appear more variable in *ser-4* or *dop-3* mutants.

Fig 1G: Although the figure legend indicate ### denotes $P < 0.001$, there is no ### label in the figure.

Fig 2: For clarity and consistency, I suggest that the drug treatments in the graph should also be labelled with "fed."

Fig 3A: I suggest replacing "control" with "WT fed" to make it clearer

Fig 4D: Indicate what ### denotes (only # or #### have been indicated).

Fig S1G-H: Indicate what purple and blue bars mean.

Fig S4F: In the legend, it should be *daf-6* not *daf-7*

Fig S6: The title does not match the text. The text in the results section states that "del-7 mutants look phenotypically wild type" which is not consistent with the figure title that "ASICs channel modify responses to DR and food smell."

Reviewer #3 (Remarks to the Author):

It was previously reported that an amine receptor antagonist mianserin extends lifespan of *C. elegans* and this is related to dietary restriction. However, the mechanisms by which food is perceived and signaled to extend lifespan was not well-understood. In this paper, the authors reveal a mechanism for lifespan extension by dietary restriction in *C. elegans*. Extensive experiments show that perception of food odor signals through serotonin and dopamine receptors to inhibit expression of the *fmo-2* gene in intestine to limit longevity. Compounds like mianserin and thioridazine, which affects aminergic signaling, increase *fmo-2* expression thereby increasing lifespan. The finding is important in that it reveals a mechanism for the most well-studied longevity intervention, dietary restriction, and suggests that modification of aminergic signaling with these drugs could produce the benefits of dietary restriction. The paper is well-written and methods and statistical analyses used in this study in general is appropriate. However, there are some issues that needs to be addressed before publication, which I listed below.

Major points:

1. The results show that perception of food odor plays an important role in the regulation of lifespan by food. And this is one of the major findings of this paper. However, the mechanisms by which odor affects amine signaling is not addressed. From the results of *daf-6*, authors conclude that "a non-canonical sensory neurons playing a role" (line 169). However, it is reported that *daf-6* mutants respond to volatile odorants (Bargmann et al., 1993, Cell 74, 515-527) and maybe odor-sensing neurons are functional in *daf-6*. Therefore, for this conclusion, involvement of canonical odor sensory neurons (*AWA*, *AWB*, *AWC*) should be tested using odor sensation mutants or (genetic) ablations of these cells.

2. Serotonin and dopamine working downstream of odor perception to regulate *fmo-2* expression is another major finding of this paper. Authors conclude that serotonin and dopamine are epistatic to each other from the results in Figure 4. This is based on the assumption that mianserin binds to *SER-4* and thioridazine binds to *DOP-3*. Petrascheck et al. (2007) previously showed that *SER-4* is inhibited by mianserin in cultured cells. However, it is unknown if thioridazine inhibits *C. elegans* *DOP-3*, and it is unknown if mianserin inhibits *DOP-3*. Pharmacological properties of invertebrate neurotransmitter receptors are sometimes quite different from mammalian counterparts and some compounds does not always bind to homologous receptors with a similar affinity. Therefore, these compounds could be working on other receptors. These issues should be carefully discussed or evidence for thioridazine antagonizing *dop-3* but not *ser-4* and mianserin antagonizing *ser-4* but not *dop-3* should be shown. Furthermore, it is unclear from the results whether mianserin and thioridazine are working as receptor agonists or antagonists and whether activation of serotonin/dopamine signaling leads to activation or suppression of *fmo-2* expression. For example, if thioridazine is inhibiting *DOP-3*, *dop-3* knockdown animals should have increased *fmo-2* expression in the fed condition and have longer lifespan than wildtype animals (because *dop-3* knockdown animals would be similar to thioridazine treated animals). But *dop-3* mutants seem to have normal *fmo-2* expression and lifespan. *ser-4* and *tph-1* knockdown animals should also have increased *fmo-2* expression and lifespan if mianserin is an antagonist. *tph-1* and *ser-4* exhibit increased lifespan but not increased *fmo-2* expression (except in Fig S8B). Figure 4D, which shows DR-mediated *fmo-2* expression is decreased in *ser-4* and rescued by *ser-4* expression, also suggests that *ser-4* increases *fmo-2* expression. Maybe I am missing something, please clarify by providing further explanation or address this issue through additional experiments. In addition, Petrascheck et al. showed that the *SER-3* octopamine receptor also works in the mianserin-mediated longevity regulation. Involvement of octopamine signaling and possible interaction among serotonin, dopamine and octopamine signaling in this regulation should at least be discussed, given that octopamine has been shown to work downstream of both serotonin and dopamine in different contexts. (Testing *ser-3* and addressing relationship among amines would further increase the value of this paper, but this may be outside of the scope of this paper.)

Minor points:

1. Line 28-30. "We further identify an enteric neuron in this pathway that signals through the serotonin receptor 5-HT1A/*ser-4* and dopamine receptor *DRD2/dop-3*." "an enteric neuron" here refers to the NSM neuron. There is no result that clearly shows that

NSM neurons signal through dop-3. This sentence should be changed.

2. Line 57-59. "Nematode flavin-containing monooxygenase-2 (fmo-2) is necessary and sufficient to increase health and longevity downstream of DR." A reference should be added.

3. Line 89. "Table S2". There seems to be no table in this manuscript.

4. Supplemental document Line 161. "compared to DR (G)". Shouldn't it be (F)?

5. Figure S1G and H. Please explain why some boxes are purple and others are blue.

6. Figure S1C and D. The figure legend indicates that D is quantification of C. But +ethyl acetate and + 1-nonanol is similarly dark in Fed condition of C. Please clarify.

7. Figure S2B. The fluorescence is higher in fed condition than in DR condition, which is different from other results. Please clarify.

8. Some graphs are labelled in a confusing way, it is especially hard to know whether the worms were fed or DR. For example, Figure 2D is labelled, "Fed", "DR", "mianserin", "thioridazine", and "trifluoperazine". It is difficult to tell whether these drug treatments were done with Fed or DR condition. The second box (in blue) in Figure 2E seems to be the result of DR but it is difficult to tell from the way the panel is labelled. It would be helpful if "Fed" or "DR" is labelled clearly in all the figures.

9. Line 115 "tetracycline" should be "tetracyclic".

10. Figure S3D. 1st and 2nd boxes are without mianserin and without serotonin. 5th and 6th boxes are also without mianserin and without serotonin. Shouldn't the 5th and 6th ones be with serotonin?

11. Line 120-121 "Diphenyleneiodonium chloride (DPI), an inhibitor of NADPH oxidase, acts as a positive control". Authors should explain a little more about the previous report on the effect of DPI and cite the paper.

12. Line 135-136 "Importantly, we also see that mianserin treatment combined with DR does not further extend lifespan (Figure S3J)." In Figure S3J, the lifespan of DR animals is shortened by mianserin. Authors should discuss possible reasons why the lifespan of mianserin + DR animals is not as long as the lifespan of DR animals in the text.

13. Line 166-168 " We find that not only are the ASI neurons (as measured by daf-3 and daf-7 RNAi) dispensable for food perception-mediated reduction in fmo-2 expression (Figure S4C-D),". Do daf-3 or daf-7 mutants (or RNAi) lack ASI neurons? If so, please cite the paper. If not, it is an overstatement to conclude that the ASI neurons are dispensable.

14. Line 237-238 "This result is consistent with dop-3 being required for dopaminergic induction of fmo-2.", Line 250-251 " Together, these results suggest that dopamine and serotonin signaling interactively induce fmo-2 and extend lifespan under DR." The conclusion of this paper is that dopamine and serotonin inhibit induction of fmo-2. So, this sentence should be changed. However, the result in Figure 4 does appear like dopamine and serotonin signaling are required for induction of fmo-2 (please see Major Point 2).

15. Line 248-250 "we depleted dop-3 with RNAi under DR and found that dop-3 depletion increases lifespan but is not further extended by DR (Figure 4F)." In Figure 4F, the differences among strains and conditions are small. In the Methods (supplemental document), statistical analyses for lifespan assays are described, but the results of the statistical analyses are not mentioned in the Results or labelled in the figures (not only for Figure 4 but also in other figures). The results of the statistical analyses should be mentioned.

16. Figure 5E. CEP neurons are depicted in this figure. ADE and PDE neurons are also

dopaminergic neurons and there seems to be no evidence to identify CEP as the neurons regulating fmo-2 expression from this study. The label should be changed.

17. Line 308-311. "Our experimental data in *C. elegans* support a model where the lack of an attractive (food) smell leads to a loss of serotonin release from the enteric NSM neurons and lack of serotonin binding to the SER-4/5-HT1A receptor. This in turn or in combination with other cues leads to a reduction in dopamine signaling to downstream DOP-3/DRD2 receptors." These sentences suggest that serotonin work upstream of dopamine, which I believe is not supported by the results.

18. Line 324-326 "slowing locomotion in the presence of food is thought to be distinctly regulated by pharyngeal mechanosensation leading to dopamine release while dwelling behavior is potentiated by serotonin³⁷ ." The paper by Sawin et al. does not show that "pharyngeal" sensation lead to dopamine release. "pharyngeal" should be removed.

Response to reviewer comments:

Overview:

We thank the reviewers and editors for their time and invaluable input into improving our paper. The reviewers have both recognized the value of this work and pointed out many places where additional results will strengthen or extend the implications of the work. We are pleased to report that, despite the challenges of the COVID19 pandemic, we have addressed all of the points raised by the reviewers. Overall, the substantive changes (34 new data sets and corresponding textual edits, underlined in the manuscript) we have made in response to the reviewer's suggestions and editor's comments have greatly strengthened the manuscript.

We hope you will agree that this work provides a key framework and details for the network of cells, signals and receptors that are involved in the dietary restriction food smell axis. Our results, expanded by this revision, identify many aspects of this signaling pathway in addition to three small molecules that extend lifespan within the pathway. As part of our responses to the reviewer points, our principal additions to this manuscript within the revision include: 1) identification of a sensory neuron (AWC) necessary for smell to diminish *fmo-2* induction, 2) validation that attractive smell can diminish lifespan in addition to *fmo-2* induction, 3) identification of the neuronal subtype (GABA) and plausible (RIS) *ser-4* expressing interneuron within the pathway, 4) identification of CEP neurons as the key dopaminergic neurons in this pathway, and 5) identification of octopamine signaling playing a role within the pathway. Our responses to each individual reviewer comment are below (comments in black, our responses in blue), and we thank everyone for considering our work.

Reviewer #1 (Remarks to the Author):

General Comments:

Miller et al uses *C. elegans* as a model system to study serotonergic and dopaminergic signaling in DR and life span. They propose a model in which DR antagonizes biogenic amine signaling to upregulate *fmo-2* and increase life span. The reviewer felt that the manuscript fails to establish a definitive mechanism and has built on extant understanding, making incremental progress, may not be sufficient to be considered at this stage. The manuscript may benefit for proper structuring and increased clarity.

There are some aspects that are already known in the field. Some examples are:

A) While the authors were surprised to find food smell suppressing DR life span, this has already been documented (Libert et al., 2007 doi: 10.1126/science.1136610; Smith et al., 2008, doi: 10.1186/1471-213X-8-49, Zhang et al., 2021; doi: 10.1038/s43587-021-00039-1; Park et al., 2021, doi: 10.1016/j.isci.2020.101979). It is important that these are referenced.

We apologize for the lack of clarity. We were surprised that food smell reduced induction of *fmo-2*, since the intestine should directly be able to sense food ingestion and thus we expected the *fmo-2* induction would be cell autonomous. We have added citations for missed references and note that "Zhang et al., 2021" was not online until after our submission. We have added descriptions of these previous findings in the introduction. Line 50-55.

B) Mianserin, that blocks neural signaling by the neurotransmitter serotonin, is known to increase life span in a manner similar to DR (Petrascheck et al., 2007, doi: 10.1038/nature05991)

This was cited in the manuscript originally, but we have added context to describe this finding and discuss how our findings compare to this report (lines 432-436), including that mianserin was previously reported not to increase lifespan on agar plates (line 150-151).

Specific comments pertaining to the experiments, hypothesis and interpretations:

1) Why metabolically active bacteria are required to produce these smells? How can one show that metabolically inactive bacteria do not emit the compounds that worms smell? Does the solid plate DR indeed reduce smell of bacteria or the levels of attractants?

It is extremely difficult to accurately test the levels of smell/attractants emitted by bacteria on a solid plate. We have re-worded our conclusion to state “We also find that live bacteria are required to abrogate *fmo-2* induction, as the presence of bacteria killed with 0.5% paraformaldehyde does not prevent DR from inducing *fmo-2* expression (Supplementary Fig. S1A-B).” Line 88-90

2) It appears that the average life span of wild type worms is in the range of 25 days where as *C. elegans* typically have an average life span of less than 20 days. Is there a particular reason for this?

There are a number of plausible reasons, most prominently that we generally display data as days from egg, where many labs represent days from L4 (a ~2.5 day difference), in addition to our lab being carefully temperature controlled to 20°C so that temperature does not fluctuate inside or outside of the incubator when animals are transferred or scored. We also note that in the Interventions Testing Program in mice, Michigan mice routinely live longer than mice from Texas or Maine, so there could be effects of water or other environmental aspects that are not understood. It is our assertion that it is likely that an intervention that extends a longer control lifespan is equally or more robust than one that extends a shorter lifespan, as the shorter lifespan is more likely to have a defect that could be rescued outside of natural aging.

3) The authors mention that *fmo-2* is sufficient for life span extension. So, do compounds that induce *fmo-2* expression, like neutral and repellent compounds, increase life span?

We have completed experiments with 1 attractant, 1 neutral, and 1 repellent compound under fed and DR conditions and found that the attractant recapitulates a partial effect of food smell on DR mediated lifespan (**new Figure 1H**); neutral odorant does not affect lifespans of both fed and DR (**new Figure S1I**); while the repellent shortened fed and DR lifespans and generally appeared sick, which we interpret as it being toxic (**new Figure S1J**). Line 106-110. As many repellants are repellants because of toxicity, it is possible, perhaps likely, that a more exhaustive test of repellants at titrated doses would find a hormetic dose that increases lifespan, but this is not in the scope of this DR-focused manuscript.

4) The study is based on the hypothesis that DR decreases biogenic amine signaling leading to higher *fmo-2* and life span. Is it not possible that DR increases *fmo-2* independently while biogenic amine negatively regulates *fmo-2*? Food smell may activate biogenic amine signaling that negatively regulates *fmo-2*. If moderate activation of *fmo-2* is sufficient for life span extension, one may not see synergy between DR and the antagonist in this model.

The evidence in Figure 1, 3 and 4, supports a model where DR, smell and biogenic amine signaling are acting in the same pathway rather than parallel pathways. If the lack of food smell and perception from DR leads to decreased biogenic amine signaling that then induces *fmo-2*, this places biogenic amine signaling within the pathway. If the pathways were independent, biogenic amine antagonism would be predicted to be additive with DR, similar to the DR-independent compound DPI. While we cannot rule out that DR may have small, additional independent inputs into *fmo-2* induction, the previous knowledge that DR decreases biogenic amine signaling and our findings that *ser-4* RNAi, *dop-3* RNAi, *tph-1* knockout, and smell each abrogate *fmo-2* expression and also DR lifespan places biogenic amine signaling within the DR pathway.

5) In line 157, the authors hypothesize that DR reduces food smell. However, there is no evidence in support of this hypothesis. Can this be experimentally validated by measuring the attractant levels on DR plates?

As previously published work shows that bacteria release volatile compounds, substantially (>10 fold) decreasing the number of bacteria will in principle decrease the number of volatile compounds. We have re-worded our statement to state “relative lack of food smell”, since the >10 fold decrease in bacteria will decrease food smell, but is not measured in this work. Line 180.

6) Was RNAi efficiency checked by quantitative PCR under each condition?

In our experience, RNAi efficiency is extremely difficult to consistently measure when studying genes that are only expressed in a small subset of neurons in a worm. Therefore, we confirm as many key results as possible with knockout animals (e.g. *tph-1*, *ser-4*, *dop-3*, *ser-3*, *tbh-1*, *cat-2* etc.). We find that the knockout animals show grossly similar phenotypes as the RNAi knockdowns, confirming our results, but realize that some negative results could still be false negatives. We therefore have added this caveat to the text. (Line 70-71, supplemental text).

7) Why is *fmo-2* not induced on *tph-1* knock down when the hypothesis is that DR induces *fmo-2* by reducing biogenic amine? Similar question arises for *del-3(-)*.

These are both great questions brought up by the reviewer. We also wonder why this is the case and hypothesize that two potentially overlapping possibilities could explain this: 1) Because there are multiple sensory inputs for food perception, loss of just the food smell pathway is not sufficient to produce a meaningful output, and 2) when cells or genes are absent or highly knocked down throughout development, the signaling networks they regulate undergo rewiring to respond to cues through different pathways. We have added this to the discussion section and while it is beyond the scope of this work, it would be interesting to test using something like a degron tag to temporally knock down expression of these genes. Lines 473-482.

8) The legends of S5 D and E are probably swapped.

Yes, they were, and have been fixed. We are sorry for the mistake.

9) The conclusion of line 186 about food smell is probably not correct as no experiment with food smell was conducted under these conditions.

Thank you for this suggestion. We have completed these experiments as requested, using *tph-1* cell-specific KO and rescue in these experiments. We find rescuing *tph-1* expression in NSM neurons

recapitulates lifespan suppression under food smell (**new Figure S5H**) and blunts its effects when knocked out specifically in the NSM neurons (**new Figure S5J**). However, *tph-1* expression in ADF does not rescue lifespan suppression by food smell under DR (**new Figure S5G**) and slightly blunts food smell effects when knocked out specifically in ADF (**new Figure S5I**). These results support that NSM neurons play a primary role in both DR and food odor response while the ADF neurons may play a smaller role as well (**new Figure S5G-J**). We have included these results in the manuscript. Line 219-226.

10) Why is *fmo-2* and life span decoupled in case of *del-3(-)*?

In addition to the answer to #7 above, which is a wonderful question, we note that *fmo-2* and lifespan may be partially decoupled, although we haven't tested this directly, but do not appear to be completely decoupled in the *del-3* mutant. As shown in Fig. 3H and 3I, *fmo-2* is still induced under DR in these mutants, albeit less so, and lifespan is extended under DR in *del-3*. Similarly, *fmo-2* induction is not blunted by food smell in *del-3*, and neither is lifespan, maintaining the coupling of *fmo-2* induction and lifespan in this case.

11) The authors mention that mianserin mimics DR. Are all aspects of DR mimicked by mianserin? They need to check the metabolic reprogramming aspect of DR before making this comment. It is possible that biogenic amine signaling is an independent signaling that partially overlaps with how DR affects life span.

We have changed “Mianserin mimics DR” to “Mianserin mimics DR in *fmo-2*-mediated longevity” in line 270.

12) The authors use pan-neuronal rescue of *ser-4* for their experiment. It will be more informative to identify the specific neurons.

Thank you for this suggestion. To narrow down the list of potential neurons acting in our pathway, we rescued *ser-4* expression in distinct neuronal populations (promoters used in Table S1). We confirmed that *ser-4* knockout animals do not respond to the suppression of DR-mediated *fmo-2* induction by food smell (**new Figure 4E-F**). We find *ser-4* expressed exclusively in GABAergic neurons is sufficient to rescue food smell suppression of DR-mediated *fmo-2* induction (**new Figure 4E-F**). Importantly, rescuing *ser-4* expression in biogenic amine (Figure S7F-G) or glutamatergic neurons (Figure S7H-I) did not rescue the effects of *ser-4* KO. These results suggest one or more GABAergic neurons (plausibly but not definitively RIS as it is GABAergic and the highest *ser-4* expressing neuron) known to transcribe *ser-4* is responding to serotonin release during food perception. Line 288-296

13) While the premises on which this study is based on is tissue-to-tissue communication of food signal. The experiment with mammalian cell culture makes no sense. It raises the question why it is cell non-autonomous in worms and flies.

This is a very good question. We were very surprised by this result and speculate that it generally means that serotonin abundance frequently signals a replete nutrient environment, and even in cases where peripheral tissues are involved, the binding of serotonin to 5HT1A receptors signals can mean similar things (i.e., “times are good/food is abundant, so turn down *Fmos*”). We note that dopamine

receptors are not expressed in the cells we used, so this could be true for dopamine as well, but is not tested here.

14) The authors have used *fmo-2* tagged transgenic line for all experiments. They should back their observation up with quantitation of the *fmo-2* transcript abundance.

See supplemental figure 2.

We have measured *fmo-2* levels as requested in Figure S3I, and see comparably increased *fmo-2* levels after 8 hours thioridazine treatment, mianserin treatment, or DR.

15) Why the values of y-axis in Figure 1 for fluorescence measurement so varied?

Under control conditions, *fmo-2* basal expression is incredibly low and the reporter is very dim. This leads to day-to-day variation in the effect size of DR when normalized to control, as any small increase or decrease in control worms, either from their own environment or microscopic variance is amplified. To control for this, we are careful to never interpret across experiments where controls may differ subtly, and to always use appropriate controls in every single experiment.

16) Figure S2B: induction of *fmo-2* is more in DR compared to fed conditions. Then, why the mean fluorescence of *fmo-2* of the controls is more in fed condition while reduced in DR?

Thank you for catching this. We apologize, these were mislabeled. It has been fixed, and the order is “fed, fed + food smell, DR, DR + food smell” instead of the previous labeling.

17) Some statistical comparisons are missing. Eg. Fig 2B and C. between DR+smell and DR+smell+antagonist.

Thank you for pointing this out. We have added statistical comparisons between the last two groups “DR+smell” and “DR+smell+antagonist” in Fig. 2B and C and both are statistically significant.

18) A recent study, Zhang et al 2021 shows the involvement of ADF neurons in food odour-dependent suppression of DR life span. The authors need to discuss why their results are different from the published one. Have the authors checked the involvement of this neuron in DR+food smell condition?

Please see the response to #9 above, which asked a similar question. In brief, we completed these experiments and NSM is clearly important for food smell effects and ADF may play a smaller role as well (new Figure S5G-J). We believe our results are cleaner because we both add and subtract *tph-1* in these neurons and only affect *tph-1* expression, whereas Zhang et al uses tetanus toxin, which affects all signaling from the neurons in question, not just serotonin. We have also discussed the different finding between the two studies in the discussion section. Line 440-448.

19) Figure 4B: *ser-4* receptors are required in neurons to exhibit DR mediated *fmo-2* induction. Have you checked whether *ser-4* receptors are also involved in *fmo-2* modulation under DR+smell scenario?

Please see the response to #12, above, repeated here. We have now included rescue of *ser-4* in neuronal populations under food smell and DR conditions. To narrow down the list of potential neurons acting in our pathway, we rescued *ser-4* expression in distinct neuronal populations (promoters used in Table S1). First, we confirmed that *ser-4* knockout animals do not respond to the

suppression of DR-mediated *fmo-2* induction by food smell (**new Figure 4E-F**). We find *ser-4* expressed exclusively in GABAergic neurons is sufficient to rescue food smell suppression of DR-mediated *fmo-2* induction (**new Figure 4E-F**). Importantly, rescuing *ser-4* expression in biogenic amine (Figure S7F-G) or glutamatergic neurons (Figure S7H-I) reflect similar changes in *fmo-2* induction as the global *ser-4* KO. These results suggest one or more GABAergic neurons known to transcribe *ser-4* is responding to serotonin release during food perception. Line 288-296.

20) Figure 4(E,F): shows that dopaminergic signaling is required for induction of *fmo-2* under DR. Have you checked which of the CEP, ADE or PDE dopamine synthesizing neurons is involved in smell perception during DR?

To test which dopaminergic neuron is responding to food odor, we expressed *cat-2* in CEP, ADE or PDE neurons in the *cat-2* KO strain lacking dopamine production and crossed it with *fmo-2p::mCherry* reporter strain. We find, as expected, that dopamine synthesis is required for *fmo-2* induction to be suppressed by food odor under DR (**new Figure 3J-K**) and that rescuing dopamine production in CEP neurons restores the suppression of *fmo-2* induction by food odor (**new Figure 3J-K**). However, *cat-2* rescue in ADE or PDE neurons does not consistently restore the food odor blunting of *fmo-2* induction (new Figure S6D-G). Line 244-253.

Reviewer #2 (Remarks to the Author):

This manuscript builds on prior work on the role of serotonin and dopamine in *C. elegans* aging in the context of dietary restriction. The principal areas of novelty lie in connecting dietary restriction and sensing of bacterial odors to the regulation of *fmo-2*, a flavin-containing monooxygenase, as well as in evidence that aspects of flavin-containing monooxygenase regulation are conserved in other species.

Major issues:

1. Mianserin and thioridazine induce *fmo-2* expression at 100 μ M (Figure 2A-E and 4A) but at this concentration these drugs do not have an effect on lifespan (Fig. 2F-G). Thus, this data does not support the conclusion that the induction of *fmo-2* expression by these drugs leads to lifespan extension. This issue should be addressed by examining *fmo-2* expression under concentrations of mianserin and thioridazine that extend lifespan (e.g., 25 μ M).

These initial assays were used for screening purposes and were then titrated for lifespan effects. We have now measured them at the lower concentrations of 25 μ M mianserin or 25 μ M thioridazine required for lifespan extension and show that they similarly induce *fmo-2* at these concentrations. (new Figure S3J-K, Line 151-152).

2. The roles of serotonin in dietary restriction and aging, and of dopamine in aging have been previously described. This paper has overlooked precedents in the literature that should be described:

a. *tph-1* has been previously shown to have a role in lifespan changes due to dietary restriction. Zhang et al., Nature Aging 2021 (<https://doi.org/10.1038/s43587-021-00039-1>) showed in that *tph-1* mutants under DR do not shorten lifespan in response to bacteria odor, using a similar set up as this manuscript.

b. Entchev et al., eLife 2015 (<https://doi.org/10.7554/eLife.06259>), showed that *tph-1* mutants affect lifespan under DR and that *tph-1* in NSM plays is important in this process.

c. Multiple drugs targeting HTR2A receptors (including mianserin) and multiple drugs targeting DRD2 (including thioridazine) have previously been shown to extend *C. elegans* lifespan in Ye et al., Aging Cell 2014 (<https://doi.org/10.1111/ace.12163>).

It is gratifying that the authors could reproduce results from the literature, but such prior work should be described to provide a more accurate context for the results presented.

We have described these papers (Line 50-55, line 58-60, line 150-151, and line 440-448) and discussed the caveats such as mianserin previously not extending lifespan on solid agar plates.

3. Benedetto et al., PLOS Genetics 2010 (<https://doi.org/10.1371/journal.pgen.1001084>) showed that loss of dopamine signaling via *dat-1* and *cat-2* mutations shorten lifespan, which differs from the lifespan extensions due to thioridazine treatment. Together, these apparent discrepancies suggest a more complex role for dopamine which should be discussed.

This is a good point, and dopamine is likely indeed complex. We have cited this paper in the discussion. We performed lifespans of *cat-2* mutant and do not see a difference between *cat-2* and WT in longevity, noting that we use a different *cat-2* mutant. In one replicate, *cat-2* has a slightly longer lifespan than wildtype worms (see data below). In another replicate, *cat-2* lives slightly shorter compared to the wildtype worms (see data below). We also discuss, thanks to a previous comment from reviewer 2, that dopamine antagonists have been previously found to extend lifespan (Ye et al., Aging Cell 2014). It is possible, perhaps likely, that dopamine can have both positive and negative effects on lifespan depending on the context. Line 466-471.

4. In contrast to Rhoades et al., (ref 30 in the manuscript), which describes NSM as a neuron that is activated by food ingestion, the results presented in this manuscript suggests that NSM is also important for sensing food odors (or relaying that information). Moreover *daf-6* mutants, where amphid sensory neuron function are disrupted, maintain the ability to sense food odors (presumably by NSM). This difference in the role of NSM (sensing odor versus ingestion) should be discussed. The manuscript uses the term “food perception” – this is vague and should be replaced with the type of perception (food ingestion, odor etc.) to be more precise.

We agree that food perception is vague, and have changed to clearer terms such as food ingestion or food odor throughout the text. We agree that this result is surprising, and because of this, confirmed

these results by both knocking out *tph-1* in the individual ADF and NSM neurons, and by rescuing *tph-1* in these same neurons. We now add that rescuing *tph-1* expression in NSM neurons recapitulates similar lifespan suppression under food smell (new Figure S5H) and blunts its effects when knocked out specifically in the NSM neurons (new Figure S5J). However, *tph-1* expression in ADF does not rescue lifespan suppression by food smell under DR (new Figure S5G) and slightly blunts food smell effects when knocked out specifically in ADF (new Figure S5I). These results suggest that NSM is clearly important for relaying food smell effects and ADF may play a smaller role as well (new Figure S5G-J).

As this is an important point, we have added discussion about the differences between NSM sensing food odor in this manuscript versus NSM sensing food ingestion in Rhoades et al.. For example, in Rhoades et al., *del-3* and *del-7* are required for NSM activation of post-food encounter slowing, but may not be involved in the behavioral changes from food odor (Line 448-452). Our data show that *daf-6* mutants where amphid sensory functions are disrupted maintain the ability to sense smell for *fmo-2* induction under DR. Our new data, suggested by reviewer 3 (new **Figure 3A-B**; new Figure S4G-J), show that AWC, which is not affected in *daf-6*, is important for the food smell-mediated blunting of *fmo-2* induction, consistent with AWC perceiving the smell and possibly signaling through NSM and/or CEP to release serotonin and dopamine (Line 193-200).

5. The supplementary tables were not included in the files made available to me. Thus, it was not possible to assess the statistical data such as the number of trials, no. of animals tested etc. Similarly, there was no information on the strains used in this paper, including alleles, transgenes, and number of outcrosses.

We apologize, these tables were prepared and submitted but were not transferred successfully. They are included in this revised submission. As shown in the tables, we use large animal cohorts for nearly every experiment and each experiment is completed at least 3 times for primary findings and at least 2 times for all experiments. Similarly, the information on the strains used in this paper, including alleles, transgenes, and number of outcrosses, is included in Supplementary Table 1.

Minor issues:

The paper draws conclusions by combining results from both RNAi feeding and mutant data. Results from these different types of experiments are difficult to compare because RNAi involves feeding animals a different type of bacteria (HT115) with known impact on lifespan (Maier et al., PLOS Biology 2010 <https://doi.org/10.1371/journal.pbio.1000376>; Pang & Curran, Cell Met. 2014 <https://doi.org/10.1016/j.cmet.2013.12.005>). The authors should acknowledge this technical issue as a caveat in the methods section.

We appreciate the clear and useful information for the references. We have cited these references and have added acknowledgement of this technical issue as a caveat in the methods section: "It is notable that RNAi feeding bacteria are HT115 which are reported to modify lifespan compared to OP50. This should be taken into consideration when comparing the lifespan data of RNAi feeding for gene knock down with the lifespan data of gene mutants fed on OP50." Line 71-74 (Supplemental text)

FUDR is a confounding factor in lifespan studies (Anderson et al, Mech Age Dev 2016 <https://doi.org/10.1016/j.mad.2016.01.004>). This should also be acknowledged as a caveat in the methods.

We have cited this reference, acknowledging that “It is notable that FUDR can extend lifespan and is a confounding factor contributing to lifespan extension, however, to avoid matricide under DR, we include FUDR in the lifespan plates in this study. This factor should be considered when interpreting the results.” Line 60-63 (Supplemental text). Due to “bagging” in *C. elegans*, DR studies are nearly impossible to carry out with a significant number of animals without FUDR.

It is surprising that RNAi of neuronal genes is so effective in this study, since neurons are resistant to RNAi in *C. elegans*. Can the authors briefly explain how they achieved effective knockdown (Supplementary Table 3 was not available for me, perhaps this was addressed there), or whether a slight reduction is sufficient?

The strain information is listed in Supplementary Table 1, which has been added. In some noted cases, we used TU3311, a neuronal RNAi hyperresponsive strain, and in most cases, we use multigenerational RNAi.

The text implies that *unc-13* is required only for synaptic vesicle exocytosis, which is not accurate. In fact, *unc-13* is also required for dense core vesicle exocytosis as well (Li & Kim, Wormbook 2008 <https://doi.org/10.1895/wormbook.1.142.1>)

Thank you for pointing this out, as there is some disagreement in the field on this. We have cited this reference and changed this description in the text as “Our results, knocking down *unc-13*, a gene required for both synaptic vesicle and dense core vesicle exocytosis, support short-range neurotransmitters and/or long-range neuropeptides as necessary for *fmo-2* induction (Figure S4A-B).” Line 183-185.

The discussion describes an on-off toggle state for food perception. This is not supported by the experiments. Since only 2 conditions were tested, it is not possible to know if additional intermediate responses are possible. However, many papers in the field have shown graded responses in lifespan extension under multiple levels of DR, which is not compatible with a toggle model.

We agree that we failed to adequately describe what we meant. We have now removed the On/Off toggle state discussion from the text, and modified it as “we hypothesize that the presence of food odor acts as a signal sensed by AWC neurons, which signal to NSM and CEP neurons to trigger serotonin and dopamine release, respectively (Figure 5E). The released serotonin and dopamine bind to serotonin receptor SER-4 on one or more GABAergic neurons and the dopamine receptor DOP-3 respectively to suppress *fmo-2* induction and longevity (Figure 5E). In contrast, under sDR, food signals are likely decreased to a level insufficient to excite the neural circuits involving AWC, NSM, and CEP neurons. Decreased serotonin and dopamine release under DR can be mimicked by serotonin and dopamine antagonists that induce *fmo-2* and promote longevity (Figure 5F).” Line 420-429

Line 246: Not clear what dysregulation means in this context. In Figure S7F-G, S8B-D, *fmo-2* expression does not appear more variable in *ser-4* or *dop-3* mutants.

We have removed this term from the text, since, as the reviewer correctly points out, *ser-4* and *dop-3* mutants are not more variable and largely recapitulate what is observed in RNAi experiments.

Fig 1G: Although the figure legend indicate ### denotes $P < 0.001$, there is no ### label in the figure.

We apologize, the ### legend has been removed, as it is not used in this figure.

Fig 2: For clarity and consistency, I suggest that the drug treatments in the graph should also be labelled with “fed.”

Thank you for this suggestion, we have added this label.

Fig 3A: I suggest replacing “control” with “WT fed” to make it clearer

That is a great idea. We have replaced “control” with “WT fed” in the original Figure 3A (now Figure 3C).

Fig 4D: Indicate what ### denotes (only # or #### have been indicated).

We have added “### denotes $P < 0.001$ when compared to *ser-4* (-) or *ser-4* (-) + intestinal fed (Turkey’s HSD)” in the figure legend.

Fig S1G-H: Indicate what purple and blue bars mean.

We are sorry about this confusion. We have now changed the color to blue for all the bars of “attractants”, “neutrals” and “repellants”.

Fig S4F: In the legend, it should daf-6 not daf-7

We have corrected it to state “*daf-6*”.

Fig S6: The title does not match the text. The text in the results section state that “*del-7* mutants look phenotypically wild type” which is not consistent with the figure title that “ASICs channel modify responses to DR and food smell.”

We have replaced the title of Figure S6 to “*del-7* mutants look phenotypically wild type in their induction of *fmo-2* and lifespan extension.”

Reviewer #3 (Remarks to the Author):

It was previously reported that an amine receptor antagonist mianserin extends lifespan of *C. elegans* and this is related to dietary restriction. However, the mechanisms by which food is perceived and signaled to extend lifespan was not well-understood. In this paper, the authors reveal a mechanism for lifespan extension by dietary restriction in *C. elegans*. Extensive experiments show that perception of food odor signals through serotonin and dopamine receptors to inhibit expression of the *fmo-2* gene in intestine to limit longevity. Compounds like mianserin and thioridazine, which affects aminergic signaling, increase *fmo-2* expression thereby increasing lifespan. The finding is important in that it reveals a mechanism for the most well-studied longevity intervention, dietary restriction, and suggests that modification of aminergic signaling with these drugs could produce the benefits of dietary restriction.

The paper is well-written and methods and statistical analyses used in this study in general is appropriate. However, there are some issues that needs to be addressed before publication, which I listed below.

Major points:

1. The results show that perception of food odor plays an important role in the regulation of lifespan by food. And this is one of the major findings of this paper. However, the mechanisms by which odor affects amine signaling is not addressed. From the results of *daf-6*, authors conclude that "a non-canonical sensory neurons playing a role" (line 169). However, it is reported that *daf-6* mutants respond to volatile odorants (Bargmann et al., 1993, Cell 74, 515-527) and maybe odor-sensing neurons are functional in *daf-6*. Therefore, for this conclusion, involvement of canonical odor sensory neurons (AWA, AWB, AWC) should be tested using odor sensation mutants or (genetic) ablations of these cells.

We thank the reviewer for pointing this out as this was a great idea. To test the necessity of these sensory neurons, we created individual genetic ablation strains and crossed them into our *fmo-2p::mCherry* reporter. We find knocking out AWC function abates *fmo-2* suppression in the presence of food smell during DR (**Figure 3A-B**) while AWA or AWB knockout acts similar to our control (Figure S4G-J). We note that AWA genetic ablation animals were very sick and developmentally delayed and could be involved in DR, but we did not observe involvement in this strain. This key result has been added to the manuscript. Line 193-200

2. Serotonin and dopamine working downstream of odor perception to regulate *fmo-2* expression is another major finding of this paper. Authors conclude that serotonin and dopamine are epistatic to each other from the results in Figure 4. This is based on the assumption that mianserin binds to SER-4 and thioridazine binds to DOP-3. Petrascheck et al. (2007) previously showed that SER-4 is inhibited by mianserin in cultured cells. However, it is unknown if thioridazine inhibits *C. elegans* DOP-3, and it is unknown if mianserin inhibits DOP-3. Pharmacological properties of invertebrate neurotransmitter receptors are sometimes quite different from mammalian counterparts and some compounds does not always bind to homologous receptors with a similar affinity. Therefore, these compounds could be working on other receptors. These issues should be carefully discussed or evidence for thioridazine antagonizing *dop-3* but not *ser-4* and mianserin antagonizing *ser-4* but not *dop-3* should be shown.

We completely agree that pharmacological properties of invertebrates and mammalian tissue cultured cells could be quite different. To test the specificity of mianserin and thioridazine in *C. elegans*, we measured the induction of *fmo-2* by mianserin or thioridazine in *ser-4* or *dop-3* knockout worms (Figure S8E-F). The results show that *fmo-2* induction by mianserin or thioridazine is decreased in *ser-4* or *dop-3* knockout worms. This suggests that induction of *fmo-2* by mianserin and thioridazine requires both *ser-4* and *dop-3*. Line 328-332

Furthermore, it is unclear from the results whether mianserin and thioridazine are working as receptor agonists or antagonists and whether activation of serotonin/dopamine signaling leads to activation or suppression of *fmo-2* expression. For example, if thioridazine is inhibiting DOP-3, *dop-3* knockdown animals should have increased *fmo-2* expression in the fed condition and have longer lifespan than wildtype animals (because *dop-3* knockdown animals would be similar to thioridazine treated animals). But *dop-3* mutants seem to have normal *fmo-2* expression and lifespan. *ser-4* and *tph-1* knockdown animals should also have increased *fmo-2* expression and lifespan if mianserin is an antagonist. *tph-1* and *ser-4* exhibit increased lifespan but not increased *fmo-2* expression (except in Fig S8B). Figure 4D, which shows DR-mediated *fmo-2* expression is decreased in *ser-4* and

rescued by *ser-4* expression, also suggests that *ser-4* increases *fmo-2* expression. Maybe I am missing something, please clarify by providing further explanation or address this issue through additional experiments.

The reviewer's assumptions are correct and this is a great question that is very difficult to cleanly answer. We have shown that *tph-1* knockout (Figure 3C), *ser-4* RNAi (Figure 4B), *dop-3* RNAi (Figure 4H) all increase lifespan. To verify this, we have also performed qPCR of *fmo-2* levels under *tph-1* RNAi, *ser-4* RNAi, *dop-3* RNAi strains but did not find changes in *fmo-2* levels when these genes were knocked down (new Figure S11). We also wonder why this is the case and hypothesize that two potentially overlapping possibilities could explain this: 1) Because there are multiple sensory inputs for food perception, loss of just the food smell pathway is not sufficient to produce a meaningful output, and 2) when cells or genes are absent or highly knocked down throughout development, the signaling networks they regulate undergo rewiring to respond to cues through different pathways. We have added this to the discussion section and while it is beyond the scope of this work, it would be interesting to test using something like a degron tag to temporally knock down expression of these genes and test whether this induces *fmo-2*. Lines 473-482

In addition, Petrascheck et al. showed that the SER-3 octopamine receptor also works in the mianserin-mediated longevity regulation. Involvement of octopamine signaling and possible interaction among serotonin, dopamine and octopamine signaling in this regulation should at least be discussed, given that octopamine has been shown to work downstream of both serotonin and dopamine in different contexts. (Testing *ser-3* and addressing relationship among amines would further increase the value of this paper, but this may be outside of the scope of this paper.)

As suggested, we asked whether octopamine is involved in this pathway. We tested whether mianserin or thioridazine induces *fmo-2* through *ser-3*. The results show that *fmo-2* induction by mianserin or thioridazine is decreased but not blocked in *ser-3* knockouts compared to wild type worms (Figure S8E-F). This is consistent with *ser-3* playing a role in *fmo-2* induction by mianserin and thioridazine. We interpret this as *ser-3* acting downstream of both drugs but not as the primary/only downstream receptor. This result agrees with previous reports that suggest octopamine signaling is downstream of dopamine in the food availability response and interacts with serotonin in regulating body fat and aversive behaviors. Based on this and our *ser-3* results, we tested whether octopamine signaling is involved in sensing food odor. We found that the octopamine synthesizing enzyme *tbh-1* and to a lesser extent *ser-3* knockout animals lack suppression of DR-mediated *fmo-2* induction in the presence of food odor. This suggests that octopamine is involved in this pathway but plausibly through multiple octopamine receptors (Figure S8G-H). Line 334-347

Minor points:

1. Line 28-30. "We further identify an enteric neuron in this pathway that signals through the serotonin receptor 5-HT1A/*ser-4* and dopamine receptor DRD2/*dop-3*." "an enteric neuron" here refers to the NSM neuron. There is no result that clearly shows that NSM neurons signal through *dop-3*. This sentence should be changed.

We have modified the text, based on this and our new data, to state "We further identify a chemosensory neuron that likely perceives food odor, an enteric neuron that signals through the serotonin receptor 5-HT1A/*ser-4*, and a dopaminergic neuron that signals through the dopamine receptor DRD2/*dop-3*." Line 28-31

2. Line 57-59. "Nematode flavin-containing monooxygenase-2 (fmo-2) is necessary and sufficient to increase health and longevity downstream of DR." A reference should be added.

We have added the reference. Line 69

3. Line 89. "Table S2". There seems to be no table in this manuscript.

We apologize, these tables were prepared and submitted but were not transferred successfully. They are now included.

4. Supplemental document Line 161. "compared to DR (G)". Shouldn't it be (F)?

We have now corrected it to F. Thank you for identifying this mistake. Line 199 (Supplemental text)

5. Figure S1G and H. Please explain why some boxes are purple and others are blue.

Reviewer 2 had the same question. We are sorry about this confusion. We have now changed the color to blue for all the bars of "attractants", "neutrals" and "repellants".

6. Figure S1C and D. The figure legend indicates that D is quantification of C. But +ethyl acetate and + 1-nonanol is similarly dark in Fed condition of C. Please clarify.

Thank you for catching this. We have now replaced original Figure S1C with representative images.

7. Figure S2B. The fluorescence is higher in fed condition than in DR condition, which is different from other results. Please clarify.

Reviewer 1 had the same question. These were mislabeled. It has been fixed. The order is "fed, fed + food smell, DR, DR + food smell" instead of the previous labeling "fed + food smell, DR + food smell, fed, DR".

8. Some graphs are labelled in a confusing way, it is especially hard to know whether the worms were fed or DR. For example, Figure 2D is labelled, "Fed", "DR", "mianserin", "thioridazine", and "trifluoperazine". It is difficult to tell whether these drug treatments were done with Fed or DR condition. The second box (in blue) in Figure 2E seems to be the result of DR but it is difficult to tell from the way the panel is labelled. It would be helpful if "Fed" or "DR" is labelled clearly in all the figures.

In Figure 2D, the drug treatments are in fed conditions. We have added "Fed" to the labelling of drugs. In Figure 2E, the drugs are treated under DR conditions. We have now labeled DR+drugs to replace the current labeling of a line representing DR above the drugs.

9. Line 115 "tetracycline" should be "tetracyclic".

We have made this change. Line 133

10. Figure S3D. 1st and 2nd boxes are without mianserin and without serotonin. 5th and 6th boxes are also without mianserin and without serotonin. Shouldn't the 5th and 6th ones be with serotonin?

Yes, thank you for catching this. The 5th and 6th ones are with serotonin. We have corrected this mistake.

11. Line 120-121 "Diphenyleneiodonium chloride (DPI), an inhibitor of NADPH oxidase, acts as a positive control". Authors should explain a little more about the previous report on the effect of DPI and cite the paper.

DPI is a drug previously identified by our lab to induce FMO-2 but has not yet been published. We have changed the text to reflect this "Diphenyleneiodonium chloride (DPI), an inhibitor of NADPH oxidase, which we previously identified in an unpublished screen to robustly induce *fmo-2* additively with DR, acts as a positive control". Line 139-140

12. Line 135-136 "Importantly, we also see that mianserin treatment combined with DR does not further extend lifespan (Figure S3J)." In Figure S3J, the lifespan of DR animals is shortened by mianserin. Authors should discuss possible reasons why the lifespan of mianserin + DR animals is not as long as the lifespan of DR animals in the text.

This is a good question and caused us to revisit our dataset. A majority of experiments show that mianserin + DR lifespans do not differ from DR experiments, while occasionally mianserin shortens DR lifespans somewhat. Thus we replaced the representative image with a more appropriate image. We suspect, based on the very narrow dose-range where mianserin extends lifespan, that mianserin can have some toxicity, even at doses where lifespan is extended. This toxicity is observed at doses at or higher than 50µM. In a mianserin fed environment, the moderate toxicity of the drug is not readily observed because the effects of activating the DR signaling pathway are stronger. However, in the animals already on DR, mianserin does not benefit the animal since the DR pathway is already activated, but where moderate toxicity is still evident, lifespan is shortened slightly.

13. Line 166-168 " We find that not only are the ASI neurons (as measured by *daf-3* and *daf-7* RNAi) dispensable for food perception-mediated reduction in *fmo-2* expression (Figure S4C-D)", Do *daf-3* or *daf-7* mutants (or RNAi) lack ASI neurons? If so, please cite the paper. If not, it is an overstatement to conclude that the ASI neurons are dispensable.

We apologize for the overstatement. We have changed this description to read "We find that loss of *daf-3* or *daf-7*, each of which leads to a loss in chemoreceptor signaling in the ASI neurons, did not affect the food odor-mediated reduction in *fmo-2* expression (Supplementary Fig. 4C-D)." Lines 190-192

14. Line 237-238 "This result is consistent with *dop-3* being required for dopaminergic induction of *fmo-2*.", Line 250-251 " Together, these results suggest that dopamine and serotonin signaling interactively induce *fmo-2* and extend lifespan under DR." The conclusion of this paper is that dopamine and serotonin inhibit induction of *fmo-2*. So, this sentence should be changed. However, the result in Figure 4 does appear like dopamine and serotonin signaling are required for induction of *fmo-2* (please see Major Point 2).

We have changed this sentence to "Together, these results suggest that dopamine and serotonin signaling interactively suppress *fmo-2* expression to limit lifespan when food and/or odor are present." (Line 324-326).

15. Line 248-250 "we depleted *dop-3* with RNAi under DR and found that *dop-3* depletion increases lifespan but is not further extended by DR (Figure 4F)." In Figure 4F, the differences among strains and conditions are small. In the Methods (supplemental document), statistical analyses for lifespan

assays are described, but the results of the statistical analyses are not mentioned in the Results or labelled in the figures (not only for Figure 4 but also in other figures). The results of the statistical analyses should be mentioned.

We have included statistical analyses in supplementary Table 2, which was excluded in error previously (Table S2). For example, in the original Figure 4F (now Figure 4H), the corrected p-value of “vector fed” vs. “vector DR” is 0.00000049 by log-rank analysis; the corrected p-value of “dop-3 DR” vs. “dop-3 fed” is 1 by log-rank analysis. There are more pairwise comparisons of conditions in Table S2. Due to the limited space in the figures and different statistical comparisons of multiple lifespan curves, we currently keep the complete statistics in Table S2 but not labelled in the figures.

16. Figure 5E. CEP neurons are depicted in this figure. ADE and PDE neurons are also dopaminergic neurons and there seems to be no evidence to identify CEP as the neurons regulating *fmo-2* expression from this study. The label should be changed.

To test which dopaminergic neuron is responding to food odor, we expressed *cat-2* in CEP, ADE or PDE neurons in the *cat-2* KO strain lacking dopamine production and crossed it with *fmo-2p::mCherry* reporter strain. We find, as expected, that dopamine synthesis is required for *fmo-2* induction to be suppressed by food odor under DR (**new Figure 3J-K**) and that rescuing dopamine production in CEP neurons restores the suppression of *fmo-2* induction by food odor (**new Figure 3J-K**). However, *cat-2* rescue in ADE or PDE neurons does not consistently restore the food odor blunting of *fmo-2* induction (new Figure S6D-G). Line 244-253

17. Line 308-311. "Our experimental data in *C. elegans* support a model where the lack of an attractive (food) smell leads to a loss of serotonin release from the enteric NSM neurons and lack of serotonin binding to the SER-4/5-HT1A receptor. This in turn or in combination with other cues leads to a reduction in dopamine signaling to downstream DOP-3/DRD2 receptors." These sentences suggest that serotonin work upstream of dopamine, which I believe is not supported by the results.

This has been changed to “Our experimental data in *C. elegans* support a model where dietary restriction (sDR) leads to decreased serotonin release from the enteric NSM neurons and decreased dopamine release from the CEP neurons. These decreases lead to reduced serotonin binding to the SER-4/5-HT1A receptor and a reduction in dopamine signaling to downstream DOP-3/DRD2 receptors. Subsequently, the loss of binding to SER-4 and DOP-3 causes a downstream induction of *fmo-2* and extension of lifespan.” Line 411-416

18. Line 324-326 "slowing locomotion in the presence of food is thought to be distinctly regulated by pharyngeal mechanosensation leading to dopamine release while dwelling behavior is potentiated by serotonin." The paper by Sawin et al. does not show that "pharyngeal" sensation lead to dopamine release. "pharyngeal" should be removed.

We have deleted “pharyngeal” in this sentence (Line 456).

REVIEWER COMMENTS

Reviewer #1 (Remarks to the Author):

The revision has significantly improved the quality and readability of the manuscript. Reference to the relevant literature has been incorporated. The discussion now provides a nice perspective to the work in the backdrop of extant literature. The authors have satisfactorily answered my queries. This will be an interesting study for the community to read.

I have a couple of points:

- 1) The mammalian data still does not make sense to me to be included in a manuscript that is showing cell non-autonomous regulation. It may be excluded.
- 2) If the authors are using the same controls in two different experiments, it will be better to mention that in the legend.
- 3) The life span extension in the DR experiment in 4B/4G is much lesser than that shown earlier. Is there another experiment that has greater life span extension that can be used?
- 4) Line 196, please explain the genetic ablation used.
- 5) Request the authors to place data of all repeats for life span experiments that was done for the work in supplementary info. This will help others to understand the variability within repeats and interpret the data.

Reviewer #2 (Remarks to the Author):

The revised manuscript addressed many of the issues I raised. However, there are remaining concerns about putting the results in the proper context, especially in the results section. The authors should make clear what is new and what is already in the literature that they have reproduced. There are also comments on the new data presented.

Adding context to the results:

Line 126

“Biogenic amines can regulate pro-longevity pathways and are involved in behavioral changes in response to food.”

The sentence that introduces this section reads as though biogenic amines have never been implicated in DR. It would have clearer to state that they have previously implicated in mediating DR, citing the papers (e.g., Petrascheck et al Nature 2007, Ye et al Aging Cell 2014, Entchev et al eLife 2015, Zhang et al Nature Aging 2021). This context would clarify that the novelty is in identifying these pathways as links between food odor and fmo-2 expression.

Line 203

“To better map this pathway that involves the serotonin antagonist mianserin, we next verified that the biogenic amine serotonin is involved in the DR-mediated longevity pathway.”

and Line 214

“To investigate the potential role of these neuron pairs...”

Similar to Line 126, there should be better context to indicate that serotonin (as tested using the tph-1 mutant) is known to mediate DR by acting in NSM and ADF. As it reads, it seems as though serotonin is a new player in DR and a new regulator of fmo-2 expression. tph-1 has been shown to act in DR from ADF and NSM (e.g., Petrascheck et al Nature 2007, Ye et al Aging Cell 2014, Entchev et al eLife 2015, Zhang et al Nature Aging 2021), and serotonin has been shown to regulate fmo-2 (Leiser et al. Science 2015)

Other comments on minor issues:

Line 188

The authors cite a paper indicating that skn-1 act from the ASI neurons to mediate DR.

However, instead of testing *skn-1*, they tested the *daf-7* TGF-beta pathway. The logic is therefore somewhat disjointed. This should be remedied by indicating that *daf-7* TGF-beta acts from ASI to modulate lifespan in response to DR and citing the relevant papers (Shaw et al. Current Biology 2007, Entchev et al., eLife 2015, Fletcher & Kim PLOS Genetics 2017)

Line 437

"We also demonstrate FMO-2, a monooxygenase that is sufficient to extend lifespan in worms..."

This statement not accurate. Increased *fmo-2* expression under 100µM mianserin or thioridazine is not sufficient to extend lifespan (Figure 2). Plays a role in lifespan extension would be more accurate phrasing.

Figure 5E

Given the evidence presented, it is premature to speculate that AWC somehow signals to the dopamine pathway in CEP and/or serotonin pathway in NSM. That serotonin and dopamine pathways are required for food odor mediated effects could imply that serotonin and dopamine act downstream or in parallel to AWC, and both possibilities are equally likely given that there is no data to distinguish between them. (For example, given the roles of biogenic amines in neural plasticity, what if these pathways regulate AWC's ability to detect or transmit food odor, or many other alternatives...)

Reviewer #3 (Remarks to the Author):

I have one minor concern with the response to one of the comments made by me. The followings are the comment and the response by the authors.

16. Figure 5E. CEP neurons are depicted in this figure. ADE and PDE neurons are also dopaminergic neurons and there seems to be no evidence to identify CEP as the neurons regulating *fmo-2* expression from this study. The label should be changed.

To test which dopaminergic neuron is responding to food odor, we expressed *cat-2* in CEP, ADE or PDE neurons in the *cat-2* KO strain lacking dopamine production and crossed it with *fmo-2p::mCherry* reporter strain. We find, as expected, that dopamine synthesis is required for *fmo-2* induction to be suppressed by food odor under DR (new Figure 3J-K) and that rescuing dopamine production in CEP neurons restores the suppression of *fmo-2* induction by food odor (new Figure 3J-K). However, *cat-2* rescue in ADE or PDE neurons does not consistently restore the food odor blunting of *fmo-2* induction (new Figure S6D-G). Line 244-253

According to Sup Table 1, CEP expression was driven by the *swip-10* promoter. Hardaway et al. (J Neurosci. 2015 Jun 24; 35(25): 9409–9423) showed the expression pattern of *swip-10*. They show that *swip-10* is weakly expressed in dopaminergic neurons but did not address if these were CEP. (They did show that "both the *swip-10b::GFP* and *swip-10c::GFP* fusions labeled cells that send cellular processes to the anterior sensilla adjacent to the CEP dendrites (Fig. 2C–E), resembling the morphology of glial-like neuronal support cells", suggesting that *swip-10* is expressed in CEP sheath cells, which are different from CEP neurons. Also, they did not address if *swip-10* is expressed in ADE or PDE.) A different promoter could be used to address if CEP is required for *fmo-2* regulation. Alternatively, authors may refrain from concluding that CEP neurons are the required neurons and only concluding that dopaminergic neurons are required and depict DA neurons instead of CEP in Figure 5.

Response to reviewer comments:

Overview:

We again thank the reviewers and editors for their time and invaluable input into improving our paper. We are pleased to report that we have addressed all of the points raised by the reviewers. These changes include removing the mammalian cell data, as requested by the editor, adding additional context within the results section, and tempering statements as requested by the reviewers. We believe that, in combination with the previous revision, the changes have further strengthened the manuscript to a very high level. Our responses to each individual reviewer comment are below (comments in black, our responses in blue), and we thank everyone for considering our work.

Dear Dr Leiser,

Thank you again for submitting your manuscript "Serotonin and dopamine modulate aging in response to food odor and availability" to Nature Communications. We have now received reports from 3 reviewers and, on the basis of their comments, we have decided to invite a revision of your work for further consideration in our journal. Your revision should address all the points raised by our reviewers (see their reports below). Please consider removing the mammalian cell culture work, as suggested by R#1. Although these experiments do provide some insight, the data is indeed rather preliminary.

As requested, we removed the mammalian cell culture work that was deemed too preliminary.

When resubmitting, you must provide a point-by-point response to the reviewers' comments. Please show all changes in the manuscript text file with track changes or colour highlighting. If you are unable to address specific reviewer requests or find any points invalid, please explain why in the point-by-point response.

The point-by-point response is below, and all changes within the manuscript are in red text.

REVIEWER COMMENTS

Reviewer #1 (Remarks to the Author):

The revision has significantly improved the quality and readability of the manuscript. Reference to the relevant literature has been incorporated. The discussion now provides a nice perspective to the work in the backdrop of extant literature. The authors have satisfactorily answered my queries. This will be an interesting study for the community to read.

I have a couple of points:

1) The mammalian data still does not make sense to me to be included in a manuscript that is showing cell non-autonomous regulation. It may be excluded.

As suggested by the reviewer and the editor, we have now removed Figure 5A-B and Supplementary Figure 9 along with the corresponding text in the results and discussion.

2) If the authors are using the same controls in two different experiments, it will be better to mention that in the legend.

Thank you for this suggestion. In the legend of Figure 3F, we added “Control (black) survival curve is also displayed in Supplementary Fig. 5F. Data in F and Supplementary Fig. 5F were acquired concurrently.” and “WT control (black) survival curve is also displayed in Supplementary Fig. 6A. Data in I and Supplementary Fig. 6A were acquired concurrently.” Line 267-268. Line 271-273. In the legend of Supplementary Fig. 5, we added “Control (pink) and DR (dotted pink line) survival curves in D and E are identical. Data in D and E were acquired concurrently.” Supplementary Information Line 223-224.

3) The life span extension in the DR experiment in 4B/4G is much lesser than that shown earlier. Is there another experiment that has greater life span extension that can be used?

Thank you for noticing that. We also noticed that the DR-mediated lifespan extension on HT115 RNAi food is less than DR effects on OP50 food. We speculate the reason is that HT115 has a higher caloric content than OP50 as measured by fat content of animals eating HT115 and reported by Stuhr and Curran (2020, *Communications Biology*). We do still see statistically significant lifespan extension on this food, but it may not be fully optimized in the standard 10^9 cfu/ml seeded protocol for the sDR condition. A titration to a more diluted concentration of HT115 may achieve a maximal extension for DR on HT115 RNAi food, and we will optimize this for future work.

4) Line 196, please explain the genetic ablation used.

The genetic ablation strains were described in the methods part of the supplementary information. We have now also described it briefly in the text, “we created individual genetic ablation strains by expressing the pro-apoptotic *caspase-3* gene under promoters of genes specifically expressed in AWA, AWB, or AWC neurons and crossed them into our *fmo-2p::mCherry* reporter.” Line 196-199.

5) Request the authors to place data of all repeats for life span experiments that was done for the work in supplementary info. This will help others to understand the variability within repeats and interpret the data.

All lifespan replicates are listed in Supplementary Table 2 “Main Figure statistics” and “Supplementary Figure statistics” tabs. We have now included a sentence referencing that “all lifespan replicates in this study are listed in Supplementary Table 2”. Line 86-87.

Reviewer #2 (Remarks to the Author):

The revised manuscript addressed many of the issues I raised. However, there are remaining concerns about putting the results in the proper context, especially in the results section. The authors should make clear what is new and what is already in the literature that they have reproduced. There are also comments on the new data presented.

Adding context to the results:

Line 126

“Biogenic amines can regulate pro-longevity pathways and are involved in behavioral changes in response to food.”

The sentence that introduces this section reads as though biogenic amines have never been implicated in DR. It would have clearer to state that they have previously implicated in mediating DR, citing the papers (e.g., Petrascheck et al Nature 2007, Ye et al Aging Cell 2014, Entchev et al eLife 2015, Zhang et al Nature Aging 2021). This context would clarify that the novelty is in identifying these pathways as links between food odor and fmo-2 expression.

Thank you for the suggestion. To emphasize the background knowledge that we described in the introduction part (Line 49), we have now added a sentence here and cited these papers: “Some biogenic amines have previously been reported to regulate DR mediated longevity^{8,9,17,28}.” Line 126-127.

Line 203

“To better map this pathway that involves the serotonin antagonist mianserin, we next verified that the biogenic amine serotonin is involved in the DR-mediated longevity pathway.”

We are sorry about the confusion. To emphasize the background knowledge that we described in the introduction part (Line 49-54), we have now added a sentence here and cited these papers: “Previous studies report that serotonin regulates DR and mianserin mediated longevity in liquid culture^{17,28}. To further map this pathway that involves the serotonin antagonist mianserin, we first verified that the biogenic amine serotonin is involved in the DR-mediated longevity pathway.” Line 206-209.

and Line 214

“To investigate the potential role of these neuron pairs...”

Similar to Line 126, there should be better context to indicate that serotonin (as tested using the tph-1 mutant) is known to mediate DR by acting in NSM and ADF. As it reads, it seems as though serotonin is a new player in DR and a new regulator of fmo-2 expression. tph-1 has been shown to act in DR from ADF and NSM (e.g., Petrascheck et al Nature 2007, Ye et al Aging Cell 2014, Entchev et al eLife 2015, Zhang et al Nature Aging 2021), and serotonin has been shown to regulate fmo-2 (Leiser et al. Science 2015)

We are sorry about the confusion. To emphasize the background knowledge that we described in the introduction part (Line 49-54), we have now added a sentence here and cited these papers: “ADF and NSM neurons are also reported to regulate food abundance sensing and food deprivation-mediated longevity^{8,9}.” Line 218-219.

Other comments on minor issues:

Line 188

The authors cite a paper indicating that *skn-1* act from the ASI neurons to mediate DR. However, instead of testing *skn-1*, they tested the *daf-7* TGF-beta pathway. The logic is therefore somewhat disjointed. This should be remedied by indicating that *daf-7* TGF-beta acts from ASI to modulate lifespan in response to DR and citing the relevant papers (Shaw et al. Current Biology 2007, Entchev et al., eLife 2015, Fletcher & Kim PLOS Genetics 2017)

Thank you for the suggestion. We have now added a sentence and cited these papers as suggested: “DAF-7/TGFβ produced by the ASI neurons modulates DR longevity^{8,35,36}.” Line 189-190.

Line 437

“We also demonstrate FMO-2, a monooxygenase that is sufficient to extend lifespan in worms...”

This statement not accurate. Increased *fmo-2* expression under 100μM mianserin or thioridazine is not sufficient to extend lifespan (Figure 2). Plays a role in lifespan extension would be more accurate phrasing.

This statement is accurate, from previously published work showing that overexpression of FMO-2 is sufficient to extend lifespan in worms. In the case of high dose mianserin or thioridazine, FMO-2 is likely no longer sufficient to overcome the toxicity of the drug, as evidenced by the shorter lifespans with increasing doses of the drugs. We have added the reference to support this statement. Line 421.

Figure 5E

Given the evidence presented, it is premature to speculate that AWC somehow signals to the dopamine pathway in CEP and/or serotonin pathway in NSM. That serotonin and dopamine pathways are required for food odor mediated effects could imply that serotonin and dopamine act downstream or in parallel to AWC, and both possibilities are equally likely given that there is no data to distinguish between them. (For example, given the roles of biogenic amines in neural plasticity, what if these pathways regulate AWC's ability to detect or transmit food odor, or many other alternatives...)

Thank you for the comments, we agree. We have modified the text as suggested: “In our working model, we hypothesize that the presence of food odor acts as a signal sensed by AWC neurons, which signal upstream, downstream, or in parallel with NSM and CEP neurons that release serotonin and dopamine, respectively (Figure 5C).” We also modified the model figure (Figure 5C) accordingly by deleting the arrows that indicate AWC signaling to CEP and NSM neurons. Line 403-405.

Reviewer #3 (Remarks to the Author):

I have one minor concern with the response to one of the comments made by me. The followings are the comment and the response by the authors.

16. Figure 5E. CEP neurons are depicted in this figure. ADE and PDE neurons are also dopaminergic neurons and there seems to be no evidence to identify CEP as the neurons regulating *fmo-2* expression from this study. The label should be changed. To test which dopaminergic neuron is responding to food odor, we expressed *cat-2* in CEP, ADE or PDE neurons in the *cat-2* KO strain lacking dopamine production and crossed it with *fmo-2p::mCherry* reporter strain. We find, as expected, that dopamine synthesis is required for *fmo-2* induction to be suppressed by food odor under DR (new Figure 3J-K) and that rescuing dopamine production in CEP neurons restores the suppression of *fmo-2* induction by food odor (new Figure 3J-K). However, *cat-2* rescue in ADE or PDE neurons does not consistently restore the food odor blunting of *fmo-2* induction (new Figure S6D-G). Line 244-253

According to Sup Table 1, CEP expression was driven by the *swip-10* promoter. Hardaway et al. (J Neurosci. 2015 Jun 24; 35(25): 9409–9423) showed the expression pattern of *swip-10*. They show that *swip-10* is weakly expressed in dopaminergic neurons but did not address if these were CEP. (They did show that “both the *swip-10b::GFP* and *swip-10c::GFP* fusions labeled cells that send cellular processes to the anterior sensilla adjacent to the CEP dendrites (Fig. 2C–E), resembling the morphology of glial-like neuronal support cells”, suggesting that *swip-10* is expressed in CEP sheath cells, which are different from CEP neurons. Also, they did not address if *swip-10* is expressed in ADE or PDE.) A different promoter could be used to address if CEP is required for *fmo-2* regulation. Alternatively, authors may refrain from concluding that CEP neurons are the required neurons and only concluding that dopaminergic neurons are required and depict DA neurons instead of CEP in Figure 5.

Thank you for the helpful comment. We have changed CEP neurons to dopaminergic neurons (DA) with CEP in parentheses as the most likely but not fully proven neuron in Figure 5C. We have also added “likely” to our statements surrounding CEP neurons. Line 248. Line 254-255. Line 258. Line 396. For *cat-2* rescue in dopaminergic neurons, we used *swip-10* promoter for CEP neurons, *trpa-1* promoter for PDE neurons, and *tax-2* promoter for ADE neurons. We completely agree that these promoters are not perfect for CEP/PDE/ADE specific expression. *Trpa-1* (used as PDE promoter here) is expressed in PDE neurons but no other dopaminergic neurons (Figure 1b-j, Nat Neurosci. 2007 May;10(5):568-77). *Tax-2* (used as ADE promoter here) is expressed in the AWC, AFD, ASE, ASG, ASJ, ADE, and BAG neurons (Figure 3A, Neuron.1996 Oct;17(4):695-706.). We show that *trpa-1p::CAT-2* and *tax-2p::CAT-2* cannot rescue the food odor blunting of *fmo-2* induction by *cat-2* knockout (Figure S6D-G), suggesting that PDE and ADE are unlikely to be involved. For *swip-10* used as CEP promoter here, as you suggested, Hardaway et al. (J Neurosci. 2015 Jun 24; 35(25): 9409–9423) show

that *swip-10* is expressed both in glial-like neuronal support cells adjacent to the CEP dendrites and also in three head cell bodies of DA neurons marked by *dat-1p::mCherry*, likely CEP neurons and ADE neurons from neuronal morphology. However, as mentioned by the reviewer, it is not completely clear whether *swip-10* is specific to CEP for dopaminergic neurons. Our data showing that *swip-10p::CAT-2* can restore the food odor blunting of *fmo-2* induction, while *tax-2p::CAT-2* (ADE expression) cannot, provides evidence that CEP is likely necessary for our phenotype (Figure 2C-E). However, we completely agree that additional promoters are needed to further confirm that only CEP neurons are required for food odor signaling pathway, and have thus tempered our statements and our model.

REVIEWERS' COMMENTS

Reviewer #1 (Remarks to the Author):

The authors have satisfactorily addressed all my queries.

Reviewer #2 (Remarks to the Author):

The authors have addressed all my concerns in this revision. I thank them for their efforts.

Reviewer #3 (Remarks to the Author):

Please mention in the Fig 5 legend that DA stands for dopaminergic neuron since there are DA neurons in *C. elegans* and it could be confusing.

Response to Reviewer Comments:

REVIEWERS' COMMENTS

Reviewer #1 (Remarks to the Author):

The authors have satisfactorily addressed all my queries.

Reviewer #2 (Remarks to the Author):

The authors have addressed all my concerns in this revision. I thank them for their efforts.

Reviewer #3 (Remarks to the Author):

Please mention in the Fig 5 legend that DA stands for dopaminergic neuron since there are DA neurons in *C. elegans* and it could be confusing.

Thank you for the suggestion. We have added ““DA” stands for dopaminergic neuron.” in the figure 5 legend.